# Confidence Elicitation: A New Attack Vector for Large Language Models

**Brian Formento[1,2], Chuan Sheng Foo[2,3], See-Kiong Ng[1]**

[1]Institute of Data Science, National University of Singapore
[2]Institute for Infocomm Research, A*STAR
[3]Centre for Frontier AI Research, A*STAR

`brian.formento@u.nus.edu`
`foo_chuan_sheng@i2r.a-star.edu.sg`
`seekiong@nus.edu.sg`

## Abstract

A fundamental issue in deep learning has been adversarial robustness. As these systems have scaled, such issues have persisted. Currently, large language models (LLMs) with billions of parameters suffer from adversarial attacks just like their earlier, smaller counterparts. However, the threat models have changed. Previously, having gray-box access, where input embeddings or output logits/probabilities were visible to the user, might have been reasonable. However, with the introduction of closed-source models, no information about the model is available apart from the generated output. This means that current black-box attacks can only utilize the final prediction to detect if an attack is successful. In this work, we investigate and demonstrate the potential of attack guidance, akin to using output probabilities, while having only black-box access in a classification setting. This is achieved through the ability to elicit confidence from the model. We empirically show that the elicited confidence is calibrated and not hallucinated for current LLMs. By minimizing the elicited confidence, we can therefore increase the likelihood of misclassification. Our new proposed paradigm demonstrates promising state-of-the-art results on three datasets across two models (LLaMA-3-8B-Instruct and Mistral-7B-Instruct-V0.3) when comparing our technique to existing hard-label black-box attack methods that introduce word-level substitutions. The code is publicly available at GitHub: Confidence_Elicitation_Attacks.

## 1 Introduction

Deep learning has demonstrated remarkable performance across a variety of tasks and fields, including computer vision, NLP, speech recognition, and graph representation learning. These technologies are employed for tasks such as classification, question answering, and more. However, deep learning models are known to be vulnerable to adversarial attacks (Szegedy et al., 2014). This vulnerability persists even as models have scaled up to include billions of parameters, especially in the form of LLMs.

Such vulnerabilities are particularly concerning in critical applications, such as healthcare (Savage et al., 2024), socio-technical systems and human-machine collaboration (Sale et al., 2024). For example, in healthcare, where a medical system provides a diagnosis, an attacker might introduce input perturbations, aiming to achieve a misclassification. In clinical support systems, such misclassifications can have lethal consequences (Ghaffari Laleh et al., 2022).

Providing confidence estimates through confidence elicitation, whether in a template or a free-form generation, has been shown to enhance the performance and utility of these systems. This is particularly important in domains where assessing the reliability of a model's responses is crucial for effective risk assessment, error mitigation, selective generation, and minimizing the effects of hallucinations. As a result, we can anticipate these techniques to become more widespread. Consequently, exploring whether we can strengthen adversarial perturbations using confidence estimates is an important area of research, with the aim of designing more robust systems.

Adversarial attacks can be classified primarily into three categories: **white-box** attacks, where every part of the model, including the gradients, is known at the time of the attack; **grey-box** attacks, where some information, such as input embeddings, output logits or probabilities are available, and **black-box** attacks, where no information except for the output prediction is known. Based on this classification, common types of adversarial attacks include gradient-based methods (Ebrahimi et al., 2018; Shin et al., 2020), **soft-label** attacks a form of grey-box scenario where only output probabilities are available (Jin et al., 2019; Ren et al., 2019; Li et al., 2020), and black-box **hard-label** scenarios, where only output predictions are accessible (Maheshwary et al., 2020b; Ye et al., 2022; Yu et al., 2022; Liu et al., 2023).

Despite previous work in this area, the soft-label scenario has been regarded as unrealistic because LLMs (In particular, commercially available models) are now accessed via an API that returns only the generated output, without providing the probability distributions or logits across the categories/vocabulary (Ye et al., 2022), while current hard-label approaches often require a significant number of model queries, as they follow a top-down optimization method as illustrated in Figure 2. This method involves initially over-perturbing a sample and then gradually modifying it to maintain its semantic properties. Throughout this process, the model must be queried continuously to verify whether the changes result in the new sample remaining adversarial, which can be a problem if the API is rate-limited. Furthermore, it can be argued that expecting only basic outputs is beyond the scope for modern LLMs that perform free-form generation. In fact, most previous hard-label works focused on BERT-based models rather than the new LLMs capable of performing classification in multiple ways.

We therefore investigate the realistic scenario of hard-label attacks on LLMs and examine whether some of their emergent abilities under a free-form generation setting can be leveraged to perform these attacks.

In this paper, we demonstrate that it is possible to approximate soft labels, essentially allowing for a hard-label attack with more information. This is achieved through the technique of confidence elicitation (Xiong et al., 2024), where we simply ask the model for its own uncertainty.

Our main contributions are as follows: **Novel Attack Vector**: We are the first to investigate whether confidence elicitation can be used as a potential attack vector on LLMs, while providing strong motivations for why anyone would want to take this approach. **Effective Black-Box Optimization**: We demonstrate that confidence elicitation can be used effectively as feedback in black-box optimization to generate adversarial examples. Our evaluation across three datasets and two models illustrates that black-box optimization is achievable even with imperfectly calibrated models. **State-of-the-Art Hard-Label Attack on LLMs**: Our methodology achieves state-of-the-art performance in hard-label, black-box, word-substitution-based attacks on LLMs. Compared to the current state-of-the-art hard-label optimization technique (SSPAttack), our method results in better Attack Success Rates (ASR) with fewer queries and higher semantic similarity. We also release our code[1].

## 2 RELATED WORK

### 2.1 ATTACKS ON LLMS

The traditional adversarial attack formulation involves adding a subtle perturbation $\delta$ to the original $x$ so that $x_{\text{adv}} = x + \delta$ (Szegedy et al., 2014). There are many ways to calculate $\delta$; the overarching idea is that we want to add a perturbation $\delta$ to $x$ to identify regions of high risk in the input space (Zhu et al., 2020). This, in turn, is expected to increase the output loss for a given task $t_a$.

One of the most effective ways to find regions of high input risk remains the fast gradient sign method (FGSM) (Goodfellow et al., 2014), the basic iterative method (BIM) (Kurakin et al., 2016), and projected gradient descent (PGD) (Madry et al., 2019). These methods utilize gradient information to find an optimal $\delta$ given a bound $\epsilon$ with either one or multiple iterations.

These first-order techniques model a tractable maximization operation over a non-convex loss landscape as follows:

---

[1] We release our code in a GitHub Repository (Confidence_Elicitation_Attacks)

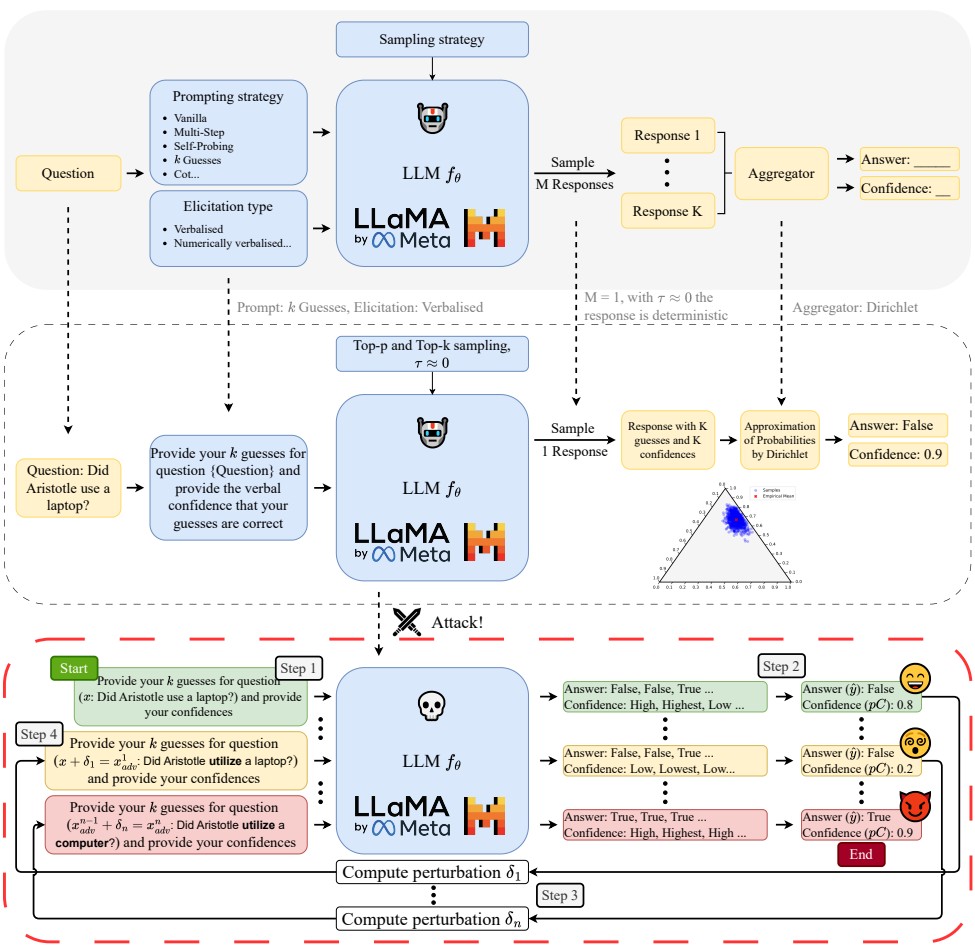

Figure 1: Confidence elicitation attack on an LLM, assuming a classification task (Start), $x$ has a ground truth $y = false$, we first perform inference and extract the model's prediction $\hat{y}$ and original elicited confidence $\mathbf{p}_{\mathcal{C}}$ (green, Step 1), we use the confidence as feedback (Step 2) to determine whether a perturbation $\delta$, modelled after a word substitution "**use**" → "**utilize**" (Step 3) added to the input leads to lower confidence (yellow, Step 4), we carry on adding $\delta$s to the input $x \to x_{adv}$ until we achieve a misclassification (red, End).

$$\rho(\theta) = E_{(x,y) \sim D} \left[ \max_{\delta \in \Delta} L(\theta, x + \delta, y) \right] \tag{1}$$

Where $\rho(\theta)$ represents the worst possible perturbation for $x$ on model $f_\theta$ with parameters $\theta$. When we maximize $L(\theta, x + \delta, y)$, we do so by using multiple $\delta$ from a set $\Delta$ of all possible perturbations given a bound $\epsilon$. This bound is often set to the $L_2$-Norm: $[|\delta|_2 \leq \epsilon]$ or $L_\infty$-Norm: $[|\delta|_\infty \leq \epsilon]$.

These techniques and their variants have found moderate success when applied to the input embedding (continuous) space as white-box attacks (Dong et al., 2021a; Zhu et al., 2020; Dong et al., 2021b). However, most language applications interact with LLMs through the token space. Although previous work has attempted to use gradient information to perform attacks in the token space (Ebrahimi et al., 2018; Zou et al., 2023; Wang et al., 2020b), the projection from a continuous to a discrete space results in high perplexity and low semantics (Zhu et al., 2024; Liu et al., 2024b).

Given the unrealistic threat model of having access to gradients and input embedding spaces, some efforts have explored adversarial attacks in the token space by perturbing at the character (Li et al., 2019; Eger et al., 2019; Formento et al., 2021; 2023), word (Jin et al., 2019; Ren et al., 2019; Li et al.,

2020; Tan et al., 2020), or sentence level (Wang et al., 2020a; Iyyer et al., 2018; Gan & Ng, 2019; Bhuiya et al., 2024). These attacks often use output probabilities as feedback while employing heuristic search techniques, such as beam search, greedy search, or particle swarm optimization (Zang et al., 2020). These methods have been exceptionally effective on BERT (Devlin et al., 2019)-based encoding models. However, widely used commercial LLMs (e.g. ChatGPT) are closed-source and logits or probabilities are not available; attacks have to operate in a purely black-box setting with only hard-predictions available as feedback (Liu et al., 2023). Recent works have explored using LLMs to red team (perform multiple black-box hard label attacks (Maheshwary et al., 2020a)) on other LLMs with moderate success (Chao et al., 2023; Xu et al., 2024). We believe the lack of feedback in the perturbation ($\delta$) optimization process is holding these attacks back.

In this paper, we propose a novel attack technique we call confidence elicitation attacks, which aim to attack models in a completely black-box setting while still utilizing feedback from the model in the form of elicitation. Our work shows promising state-of-the-art results on word substitution attacks on LLMs.

## 2.2 Confidence Elicitation

Multiple studies have explored calibration in language models. A common method, which has been thoroughly explored in previous work (Guo et al., 2017a; Jiang et al., 2021) involves using output probabilities as a proxy for confidence. This could be implemented by focusing on the first generated vector for a specific token, by adding a binary classification prediction head that utilizes the last generated token (Kadavath et al., 2022), focusing on the answer specific token or take the average of the probabilities across the whole sequence, these techniques have been classified as white-box confidence estimation. While these approaches could be effective, several challenges arise. Firstly output logits or probabilities may not be accessible, particularly with proprietary models. Secondly the likelihood of the next token primarily signifies lexical confidence and not epistemic uncertainty (Lin et al., 2022), and therefore, struggles to capture the semantic uncertainty in the entire text (Xiong et al., 2024).

As a result, previous work highlighted the need for models capable of directly expressing uncertainty in natural language in a black-box setting. Some research has explored enhancing calibration by empirically deriving confidence through repetitive model querying (Portillo Wightman et al., 2023). Alternatively, models can be prompted to express their confidence verbally, either through verbalized numerical confidence elicitation (Xiong et al., 2024) or verbal confidence elicitation (Lin et al., 2022). It has been found that some prompts can achieve reasonable uncertainty quantification, especially by querying the model twice, first for the prediction, and the second time for the uncertainty estimates (Tian et al., 2023) (Example of a prompt for confidence elicitation is in Table 6 in the Appendix).

## 3 Methodology

In this work, we assume a classification setting similar to previous work (Xu et al., 2024). Here, a model $f_\theta$ maps $f_\theta : \mathcal{X} \mapsto \mathcal{Y}$ and is used to classify a sample $x$ with ground truth label $y$. This is done by first obtaining token IDs $\mathbf{Z} \leftarrow t(x)$, where $t$ is a tokenizer, and then using such IDs to look up the corresponding embedding vectors from embedding matrix $\mathbf{E}$ so that $\mathbf{X} \leftarrow \mathbf{E}(\mathbf{Z})$. The model will then make a prediction $\hat{y} = f_\theta(\mathbf{X})$. The goal of an adversarial attack is to modify $x$ through an adversarial sample generator $g$ (which is an attack algorithm) and a perturbation $\delta$ (such as word substitutions or character insertions) to produce $x_{\text{adv}}$, such that $y \neq \hat{y}$. The adversarial sample $x_{\text{adv}}$ can be evaluated for linguistic qualitative or quantitative properties using an evaluation algorithm $d_\epsilon$.

$$\rho(\theta) = -E_{(x,y)\sim D}\left[\max_{\delta\in\Delta} C(\theta, x+\delta, y)\right] \tag{2}$$

In an adversarial settings, minimizing the confidence $C$ in the correct class $y$ can be linked to maximizing the probability of misclassification.

The approach is Equation 2 is illustrated in the dashed red box of Figure 1. As $\delta$ substitutions are added to the input sample (green), p$\mathcal{C}$ decreases (yellow) until a misclassification occurs **False** $\rightarrow$ **True** (red), at which point the confidence can be maximized.

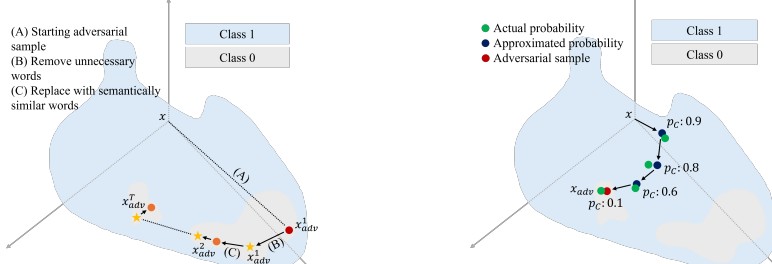

Figure 2: Confidence Elicitation Attack on an LLM: Left) SSPAttack and other previous hard-label attacks first (A) perform multiple $\delta$ word substitutions, so that a heavily perturbed sample is misclassified. Then they (B)/(C) perform further optimization to improve the adversarial sample's quality. Right) In contrast, CEAttacks take a bottom-up approach by progressively perturbing the original sample with $\delta$ word substitutions until a misclassification is achieved, using model guidance through probability approximations. The adversarial perturbation is bounded by $\epsilon$ to preserve its quality.

## 3.1 CONFIDENCE ELICITATION ATTACKS

In this section, we outline a method to attack a model by leveraging its confidence levels. Assuming an ideal scenario, where the model is black-box, calibrated and always outputs a response with it's prediction and confidence values (Ulmer et al., 2024), the process involves querying the model using a prompt that includes an output request for its answer's confidence. The response, a confidence value as a percentage between 0 and 100 or verbalised, is then subjected to string analysis. The procedure can be formulated in a generalized confidence elicitation attack, detailed in Algorithm 1.

The algorithm initiates by querying the model with an input $x$ to obtain both a prediction $\hat{y}$ and the model's $f_\theta$ confidence in that prediction, denoted as $p_{\hat{y}}$. Subsequently, a generator $g_\delta$, designed to perturb the input by a factor of $\delta$, is employed to produce an adversarial example $x_{adv}$. This adversarial input is then verified by a discriminator $d_\epsilon$ with bound $\epsilon$, which checks for adherence to specified linguistic constraints. Should the constraints not be met, $g_\delta$ is requested to generate a revised $x_{adv}$.

Once an acceptable adversarial sample is obtained, it is processed by $f_\theta$ to determine the confidence level of the adversarial example, denoted as $\mathbf{p}_\mathcal{C}$. This new confidence is then compared with the original $p_{\hat{y}}$ to decide whether to accept or reject the perturbations introduced in $x_{adv}$.

## 4 EXPERIMENTAL SETUP

We conducted our confidence elicitation attacks on Meta-Llama-3-8B-Instruct (Touvron et al., 2023) and Mistral-7B-Instruct-v0.2 (Jiang et al., 2023) while performing classification on two common datasets to evaluate adversarial robustness: *SST-2*, *AG-News* and one modern dataset: *StrategyQA* (Geva et al., 2021). We utilize the evaluation framework previously proposed in (Morris et al., 2020), where an evaluation set is perturbed, and we record the following data metrics: *Clean accuracy* (CA), *Accuracy under attack* (AUA), the *Attack success rate* (ASR), *Semantic similarity* (SemSim) based on the Universal Sentence Encoder (Cer et al., 2018). We compare the original perplexity with the new perturbed sample's perplexity. *Queries* where we subdivide this metric into two categories: *All Att Queries Avg* and *Succ Att Queries Avg*. Additionally, we track *Total Attack Time*. We compare our guided word substitution attacks, CEAttack to "Self-Fool Word Sub" from (Xu et al., 2024), TextHoaxer (Ye et al., 2022) and SSPAttack from (Liu et al., 2023). We discuss every metric and baseline in more detail in Appendix E.

## 4.1 IMPLEMENTATION

**Prompting** From Figure 1, we first initialize the model using a two-step prompting strategy. This strategy consists of an initial $k$ guess query to the model, yielding $k_{pred}$ guesses, followed by a

---

**Algorithm 1:** Confidence elicitation attack

---

**Input:** Initial input $x$, Prompt for perturbation, Vocabulary $\mathcal{V} = \{\tau_1, \tau_2, \ldots, \tau_V\}$, original
       sample confidence $p_y$

**Output:** Predicted class $\hat{y}$, adversarial sample $x_{adv}$ if conditions are met

1   Initialize generator function $g_\delta$ and input $x$.

2   **while** *not exceeded number of queries* **do**

3      **repeat**

4         $x_{adv} \leftarrow$ Output from $g_\delta$

5      **until** $d_\epsilon(x_{adv})$ *is true*

6      Compute prediction and confidence: $\hat{y}, \mathbf{p}_\mathcal{C} \leftarrow f_\theta(x_{\text{adv}})$      `// Assumes `$f_\theta$` returns`
        `prediction and calibrated confidence`

7      **if** $\mathbf{p}_\mathcal{C} < p_y$ **then**

8         Substitute $x$ with $x_{adv}$

9         $p_y \leftarrow \mathbf{p}_\mathcal{C}$         `// Update previous confidence to current`

10      **else if** $\mathbf{p}_\mathcal{C} \geq p_y$ **then**

11         Perform an alternative perturbation or adjustment.

12         $x \leftarrow$ Some function of $x_{adv}$ that modifies or reverts changes

13         $p_y \leftarrow \mathbf{p}_\mathcal{C}$         `// Optionally update the reference confidence`

---

second query to the model to obtain verbalized confidence levels for these $k$ guesses $k_{conf}$. The confidence levels used are 'Highest', 'High', 'Medium', 'Low', and 'Lowest'. This technique has been demonstrated to be effective in previous work (Tian et al., 2023; Lin et al., 2022). In our experiments we set $k$ to 20 for *SST2* and *AG-News* and $k$ to 6 for *StrategyQA*.

**Model**   Model-wise, previous confidence elicitation works (Xiong et al., 2024) use model settings commonly found in generative tasks where the model samples from the top-$k$ (top-$k = 40$) most probable next tokens, applies top-$p$ ($p = 0.92$) nucleus sampling, which only considers next tokens with high probability, and uses a temperature setting of $\tau = 0.7$. This setup naturally introduces some randomness to the model, whose behavior is still not fully understood in an adversarial setting, as highlighted in previous work (Huang et al., 2023; Zhao et al., 2024; Russinovich et al., 2024). We keep all the settings consistent with previous work, but set $\tau \approx 0$, which follows previous work related to adversarial evaluation (Xu et al., 2024).

**Dirichlet aggregation**   The differences among each of the $k_{pred}$ and $k_{conf}$ can be viewed as a form of epistemic uncertainty. To model our confidence thresholds using these $k_{pred}$ and $k_{conf}$, we employ a Dirichlet distribution. First, we construct an $\alpha$ vector where each prediction in $k_{pred}$ is assigned a value of 1. For each class, we then add the following foundational values from $k_{conf}$ : 'Highest' = 5, 'High' = 4, 'Medium' = 3, 'Low' = 2, and 'Lowest' = 1. Additionally, an $\alpha_0$ with a value of 1 is included. With this $\alpha$-vector, $\alpha = [\alpha_1, \alpha_2, \ldots, \alpha_C]$, where $C$ is the number of expected classes. The mean (expectation) of the Dirichlet distribution is given by $\mu_c = \frac{\alpha_c}{\sum_j \alpha_j}$. The mean is taken as the probability of the classifier, where the classifier uses the argmax of these probabilities for classification.

**Adversarial setup**   Given a sample $x$, we first extract an ordered subset of words $W \subset x$ to perturb randomly, with the size of $W$ capped at $|W| = 5$. For each word $w \in W$, we obtain a set $S$ of synonyms sourced from Counter-fitted embeddings (Mrkšić et al., 2016). We introduce a perturbation $\delta$ in $x$ by performing a word substitution, replacing $w$ with a synonym $s$ from $S$. For each synonym replacement, we generate a transformation. The list of all such transformations is denoted as $T_{x_{w \leftarrow S}}$, where $x_{w \leftarrow S}$ represents the original sample $x$ with the word $w$ systematically replaced by all possible synonyms in $S$. Formally, for each $w \in W$ we first identify the set $S = \{s_1, s_2, \ldots, s_n\}$, where $S$ is the set of synonyms for $w$ obtained from counter-fitted word embeddings and then perform substitutions to create transformed samples $\{x_{w \leftarrow s_1}, x_{w \leftarrow s_2}, \ldots, x_{w \leftarrow s_n}\}$. This yields a set of transformed samples $T_{x_{w \leftarrow S}} = \{x_{w \leftarrow s} \mid s \in S\}$ where we evaluate each $x_{adv} \in T_{x_{w \leftarrow S}}$ to determine if it successfully achieves a drop in confidence in the prediction. Querying the model with

$x_{adv}$ produces new $k'_{pred}$ and $k'_{conf}$, resulting in a new $\alpha'$-vector. With the new $\alpha'$-vector derived from $x_{adv}$, the new mean $\mu'$ will be $\mu'_c = \frac{\alpha'_c}{\sum_j \alpha'_j}$.

We aim to induce misclassification by minimizing the probability of the current class:

$$\mathbb{E}_{(x,y)\sim D}\left[\max_{\delta \in \Delta, d_\epsilon(x,x+\delta)\leq\epsilon} -(\mu'_y)\right] \qquad (3)$$

Where $\delta \in \Delta$ represents the set of perturbations given $|W|$ and $|S|$. For our problem, we aim to minimize the current class probability based on the Dirichlet mean, $\mu$. If a transformation $x_{adv}$ succeeds in lowering the confidence level of the model's output, we retain that word substitution. Otherwise, we proceed to the next word in $W$. This iterative process is similar to a hill-climbing greedy algorithm, where we aim to perturb the sample $x$ iteratively to achieve the desired reduction in model confidence.

## 5 RESULTS

### 5.1 MODEL CALIBRATION

By calculating the Expected Calibration Error (ECE) (Guo et al., 2017b), the Area Under the Receiver Operating Characteristic Curve (AUROC), and plotting reliability diagrams (Xiong et al., 2024), we can evaluate how well a model is calibrated for confidence elicitation. ECE is a metric used to assess how well a model's confidence estimates align with the actual probabilities of outcomes being correct. For example, it helps evaluate how accurately a model's predicted confidence (e.g., 'I'm 80% sure this is correct') matches reality. This assessment is averaged across 500 examples. A thorough explanation of ECE is provided in Appendix E.4.

We first demonstrate that LLama3 is well-calibrated for the SST2, AG-News, and StrategyQA tasks, as illustrated in Table 1 and Figure 3. In contrast, Mistral-V0.3 is reasonably well-calibrated for these same tasks, as shown in Table 1 and Figure 6 in Appendix F.3. The Expected Calibration Error (ECE) is low across all our tests. Additionally, all tests exhibit a high AUROC. Furthermore, for SST2 and StrategyQA with LLama3, the reliability plots are close to the expected diagonal, indicating that the model performs as expected on these tasks and demonstrates some awareness of its own uncertainty in the answers. Consequently, by minimizing confidence, we anticipate an increased likelihood of misclassification. These tests primarily highlight that well-calibrated models, such as LLama3 for SST2 and StrategyQA, already exist. This capability is likely to improve further in the future, as models have been shown to develop emergent abilities with increased scale, thereby making confidence elicitation attacks more powerful.

| Calibration of verbal confidence elicitation | | | | | |
|---|---|---|---|---|---|
| **Model** | **Dataset** | **Avg ECE ↓** | **AUROC ↑** | **AUPRC Positive ↑** | **AUPRC Negative ↑** |
| LLaMa-3-8B Instruct | SST2 | 0.1264 | 0.9696 | 0.9730 | 0.9678 |
| | AG-News | 0.1376 | 0.9293 | - | - |
| | StrategyQA | 0.0492 | 0.6607 | 0.6212 | 0.6863 |
| Mistral-7B Instruct-v0.3 | SST2 | 0.1542 | 0.9537 | 0.9616 | 0.9343 |
| | AG-News | 0.1216 | 0.8826 | - | - |
| | StrategyQA | 0.1295 | 0.6358 | 0.6421 | 0.6185 |

Table 1: Expected Calibration Error (ECE) and the Area Under Receiver Operating Characteristic (AUROC) of models performing zero shot classification on SST2, AG-News and StrategyQA.

### 5.2 CONFIDENCE ELICITATION ATTACK RESULTS

When feedback from the model is provided through confidence elicitation, an attack algorithm can identify approximated input perturbations that minimize the elicited confidence, thereby increasing the likelihood of misclassification. Table 2 demonstrates that our Confidence Elicitation Attack (CEAttack) achieves a higher attack success rate "ASR" for both LLama3 and Mistral across all datasets compared to "Self-Fool Word Sub" which performs no optimization, and SSPAttack/TextHoaxer which perform optimization on hard labels.

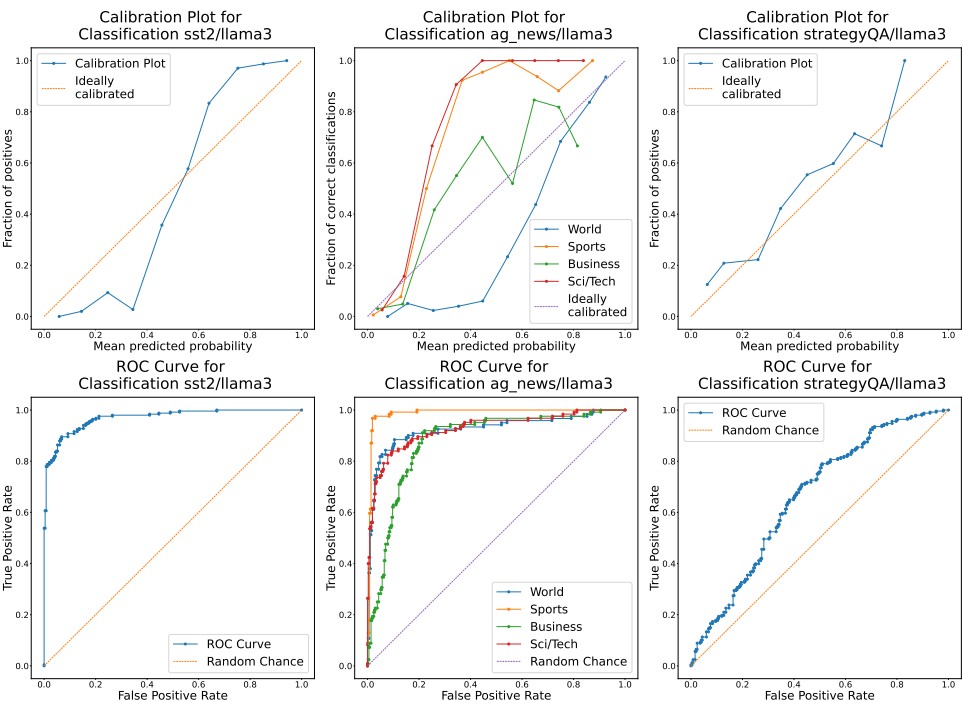

Figure 3: Reliability plots. Top) We show the SST2, AG-News and StrategyQA on LLama 3 8B Instruct calibration plots. Bottom) The ROC curves. The diagonal line is the optimal calibration.

| | | CA [%] ↑ | AUA [%] ↓ | | | | ASR [%] ↑ | | | |
|---|---|---|---|---|---|---|---|---|---|---|
| | | **Attack Performance Tests** | | | | | | | | |
| Model | Dataset | Vanilla | Self-Fool Word Sub | Text Hoaxer | SSP Attack | CE Attack | Self-Fool Word Sub | Text Hoaxer | SSP Attack | CE Attack |
| LLaMa-3-8B Instruct | SST2 | 90.56±0.14 | 88.35 | 82.93 | 81.93 | **72.69** | 2.22 | 8.43 | 9.73 | **19.73** |
| | AG-News | 61.62±0.38 | 61.17 | 49.3 | 45.27 | **43.06** | 0.33 | 19.41 | 26.71 | **30.74** |
| | StrategyQA | 60.22±0.17 | 59.52 | 45.29 | 42.28 | **32.67** | 1.66 | 24.67 | 29.67 | **45.67** |
| Mistral-7B Instruct-v0.3 | SST2 | 87.87±0.39 | 84.73 | 74.27 | 75.31 | **71.76** | 3.57 | 16.08 | 14.08 | **17.94** |
| | AG-News | 65.99±0.27 | - | 48.69 | 52.48 | **40.82** | - | 26.43 | 20.0 | **38.33** |
| | StrategyQA | 59.92±0.32 | 59.61 | 44.33 | 41.13 | **36.21** | 1.22 | 26.23 | 30.99 | **39.26** |

Table 2: Results of performing Confidence Elicitation Attacks. Numbers in **bold** are the best results

Having a high threshold $\epsilon$ allows only high-quality, label-preserving perturbations to be kept. The successful perturbations in our study all have an angular semantic similarity of at least $\epsilon = 0.84$, which is a common threshold used in previous works (Jin et al., 2019). In practice, any successful perturbation that changes the prediction while being above this threshold is deemed a successful attack. However, we find that our technique is also at times better at preserving quality when compared to the alternatives, as shown by the high "SemSim" in Table 3. This is likely due to the algorithm not having to change more words than absolutely necessary to achieve a successful perturbation. We present a detailed analysis of one qualitative example in Table 5. Additionally, a further discussion on sample quality, along with multiple qualitative examples, can be found in Appendix I and Appendix J.

| | | SemSim ↑ | | | | Original Perplexity ↓ | | | | After-Attack Perplexity ↓ | | | |
|---|---|---|---|---|---|---|---|---|---|---|---|---|---|
| | | **Quality Tests** | | | | | | | | | | | |
| Model | Dataset | Self-Fool Word Sub | Text Hoaxer | SSP Attack | CE Attack | Self-Fool Word Sub | Text Hoaxer | SSP Attack | CE Attack | Self-Fool Word Sub | Text Hoaxer | SSP Attack | CE Attack |
| LLaMa-3-8B Instruct | SST2 | 0.87 | 0.89 | 0.87 | 0.88 | 73.75 | 76.51 | 69.04 | 69.81 | 82.95 | 113.0 | 143.81 | 111.16 |
| | AG-News | 0.86 | 0.94 | 0.88 | 0.93 | 354.12 | 78.62 | 66.31 | 72.01 | 320.06 | 99.02 | 193.16 | 98.9 |
| | StrategyQA | 0.87 | 0.89 | 0.89 | 0.89 | 281.38 | 104.83 | 115.63 | 105.42 | 220.73 | 182.15 | 232.31 | 206.23 |
| Mistral-7B Instruct-v0.3 | SST2 | 0.87 | 0.9 | 0.87 | 0.88 | 79.06 | 63.03 | 63.44 | 61.68 | 91.85 | 85.27 | 118.67 | 95.85 |
| | AG-News | - | 0.94 | 0.88 | 0.93 | - | 86.47 | 74.76 | 73.2 | - | 103.25 | 188.83 | 97.19 |
| | StrategyQA | 0.89 | 0.9 | 0.89 | 0.9 | 74.04 | 85.2 | 95.43 | 97.3 | 93.57 | 140.08 | 195.33 | 177.94 |

Table 3: Quality results of performing Confidence Elicitation Attacks. Only successful perturbations are considered.

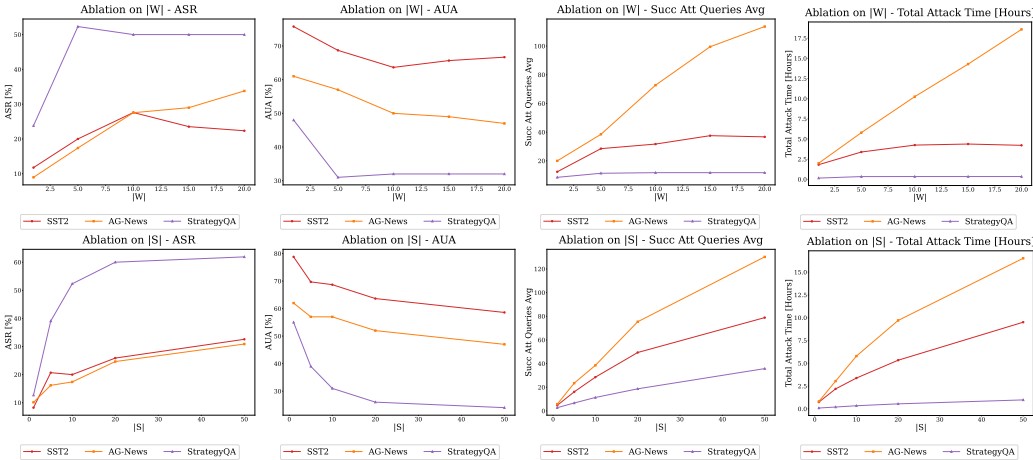

Figure 4: Ablation study on $|W|$ and $|S|$

Because CEAttack's optimization path is more direct compared to SSPAttack as illustrated in Figure 2, the algorithm doesn't need to explore a large section of the manifold. Instead, by approximating the probabilities, it finds regions of high risk closest to the original input, drastically cutting the number of required queries and optimization time. This efficiency is demonstrated in the columns "Succ Att Queries Avg" which only records the number of queries for the successful attacks, and "Total Attack Time" in Table 4.

| | | Efficiency Test | | | | | | | | | | | |
| | | All Att Queries Avg ↓ | | | | Succ Att Queries Avg ↓ | | | | Total Attack Time [HHH:MM:SS] ↓ | | | |
| Model | Dataset | Self-Fool Word Sub | Text Hoaxer | SSP Attack | CE Attack | Self-Fool Word Sub | Text Hoaxer | SSP Attack | CE Attack | Self-Fool Word Sub | Text Hoaxer | SSP Attack | CE Attack |
| LLaMa-3-8B Instruct | SST2 | 20.96 | 24.97 | 11.11 | 21.81 | na | 171.31 | 82.95 | **25.60** | 001:45:58 | 006:28:54 | 023:12:58 | 017:30:57 |
| | AG-News | 21.66 | 24.18 | 43.46 | 42.88 | na | 100.49 | 152.85 | **42.36** | 001:42:01 | 004:33:43 | 059:46:06 | 024:31:58 |
| | StrategyQA | 22.23 | 19.24 | 8.03 | 8.5 | na | 51.71 | 19.76 | **10.95** | 000:44:37 | 000:49:09 | 001:22:34 | 001:25:34 |
| Mistral-7B Instruct-v0.3 | SST2 | 20.5 | 38.88 | 13.28 | 23.29 | na | 183.6 | 73.49 | **24.54** | 001:22:23 | 007:03:41 | 023:52:30 | 017:13:44 |
| | AG-News | - | 23.96 | 34.76 | 42.84 | - | 76.71 | 158.66 | **42.66** | - | 003:43:41 | 045:50:13 | 017:16:52 |
| | StrategyQA | 20.86 | 16.66 | 8.74 | 8.71 | na | 45.71 | 21.32 | **11.37** | 000:34:41 | 000:55:14 | 001:38:57 | 001:43:48 |

Table 4: Efficiency results of performing Confidence Elicitation Attacks.

All figures in 4 present ablations for the maximum number of possible word substitutions per sample $|W|$ and the number of possible synonym embeddings per word $|S|$. The results indicate that as both parameters increase, the attack success rate also increases, provided there are enough words to perturb. The trend varies across datasets due to differences in the average number of words per example. In our evaluation, AG-News has the longest examples, resulting in the scaling of $|W|$ and $|S|$ having the most significant impact.

# 6 ANALYSING ATTACK PATHING

Our attack framework, when compared to previous work on adversarial attacks on LLMs, allows us to track how perturbations affect the model's output state. Despite being approximate, it provides valuable insights. In the experiment illustrated in Table 5 and Figure 5, we allow the search algorithm to perturb a sample beyond the boundary. Table 5 highlights an example where the prediction sentiment shifts from positive to negative after just two word substitutions. As the sample undergoes word substitutions, the probability of the wrong class increases, yet it retains its positive label over five different substitutions. As words are substituted, the empirical mean from the Dirichlet distribution moves from the positive region (top-left plot) to the most negative region (bottom-right plot) in Figure 5. The final sample is approximately 97% correlated with being negative. This approximate information would not be available in a hard label attack scenario, making it difficult to detect the confidence level of the new adversarial example. We observe this behavior across multiple examples.

| Example Analysis | | | | |
| --- | --- | --- | --- | --- |
| SST2 (Sentiment Classification) LLaMa-3-8B-Instruct | | | | |
| Technique | Sample | Perturbed Words | Prediction | Empirical Mean (Score) |
| Original | although laced with humor and a few fanciful touches, the film is a refreshingly serious look at young women. | 0 | Positive | 0.13 |
| CEAttack (Ours) | although laced with humor and a few fanciful touches, the film is a **blithely** serious look at young women. | 1 | Positive | 0.33 |
| | although laced with humor and a few fanciful touches, the film is a **blithely** serious **heed** at young women. | 2 | Negative | 0.78 |
| | although laced with humor and a few **awesome** touches, the film is a **blithely** serious **heed** at young women. | 3 | Negative | 0.94 |
| | although laced with **fun** and a few **awesome** touches, the film is a **blithely** serious **heed** at young women. | 4 | Negative | 0.97 |
| | although laced with **fun** and a few **awesome** touches, the film is a **blithely deeply heed** at young women. | 5 | Negative | 0.97 |

Table 5: Example of a sample being progressively perturbed, the Dirichlet distributions of this process in Figure 5. Perturbed words are in **bold**.

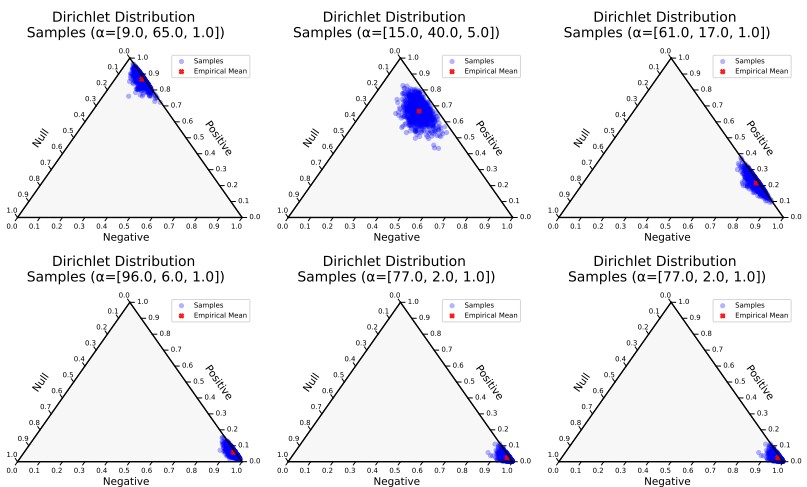

Figure 5: Ternary plots highlighting the attack path for the example in Table 5. As the algorithm adds adversarial word substitutions the model's predictions and associated confidences to such predictions change leading to a different Dirichlet distribution profile

# 7 CONCLUSION

In this work, we demonstrated that elicited confidence can serve as a feedback mechanism for identifying input perturbations. This feedback enables us to craft stronger adversarial samples. We believe this mechanism is agnostic to the type of input perturbation, the search algorithm, and the quantitative or qualitative bounds of $\epsilon$. Our word substitution attack can be tracked through substitution steps to observe how confidence diminishes and eventually alters the prediction. We achieve all of this within a fully black-box threat model, and for the first time, point out that confidence elicitation may be at odds with robustness. Our results suggest the potential for confidence elicitation to enhance jailbreaks. For example, it may enable current multi-turn dialog jailbreaks (Chao et al., 2023; Mehrotra et al., 2024) to query the probability of the model's answers and use this information as feedback. Another promising direction is to investigate the susceptibility of token-wise confidence elicitation to input perturbations, and whether it is possible to control or influence the model's token selection process. Additionally, it is worth exploring how effective confidence elicitation attacks are on generative tasks such as free-form question answering (Joshi et al., 2017), and reasoning (Gan et al., 2024; Ma et al., 2024; Bhuiya et al., 2024; Wang et al., 2024) given that confidence elicitation has also proven to be a reliable and calibrated measure of uncertainty in generative tasks (Liu et al., 2024a; Chaudhry et al., 2024). We hope that our attack, evaluation, insights, and open-source code can assist researchers in identifying sample perturbations, exploring their model's confidence elicitation behavior, and, more broadly, advancing adversarial robustness research.

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

## A   ETHICS STATEMENT

This research was conducted in accordance with the ACM Code of Ethics. Although our technique may be used to bypass refusal mechanisms, we propose in the discussion section various ways to defend against this.

## B   FUTURE WORK

### B.1   DEFENSE DISCUSSION

The main challenge with defending against this issue arises from an active push within the community to make confidence elicitation an integral part of LLMs' behavior. We believe this is an interesting emergent behavior and do not think the community should halt these efforts. Therefore, simply blocking models from performing confidence elicitation or impairing their ability by adding noise or deliberately making them uncalibrated may not be a viable option (Raina et al., 2024). Ultimately, we concluded that confidence elicitation may be at odds with robustness. However, we have identified two potential directions that the community may find worth exploring:

### B.1.1   ADVERSARIAL TRAINING / ADVERSARIAL DATA AUGMENTATION

Can the generated adversarial inputs be reintroduced into the training process? This opens up opportunities for confidence elicitation adversarial training, aiming to both enhance robustness against

input perturbation and potentially improve calibration. This would adhere to the traditional adversarial training min-max formulation (Yu et al., 2024; Xhonneux et al., 2024; Ji et al., 2024; Formento et al., 2024).

As we incorporate perturbations into the input that alter predictions during instruction fine-tuning, we can loop these samples back into the training process, following a black-box adversarial setup. Adversarial training could potentially be done in a white-box setting by perturbing the input embeddings then checking how the confidence elicitation behavior changes.

Alternatively, a simpler solution involves generating the data first and then using it for further fine-tuning (adversarial augmentation).

### B.1.2 DEFENSE BY INTENT

In this case, we aim to protect the system by analyzing the use cases of confidence elicitation with a rule-based defense approach.

- Is the user performing the same query multiple times with small semantic similarities between queries, likely adding minor input perturbations?

- Is the user explicitly asking for confidence elicitation? This could be implemented as a classifier; if yes, it indicates a desire for confidence elicitation.

- Are confidence elicitation values on some tokens decreasing over time? This could suggest some form of optimization in progress.

### B.2 APPLICATIONS OF CONFIDENCE ELICITATION ATTACKS

As briefly introduced in the paper, we target classification settings where either confidence is provided as a main component of the system as feedback or where the attacker can perform confidence elicitation in a free-form generation setting. Exploring this approach using actual medical datasets would be an interesting direction.

Although chatbot jailbreaks can fundamentally differ from adversarial attacks, there are precedents where adversarial techniques have been employed to craft jailbreaks. For example, variants of the HotFlip (Ebrahimi et al., 2018) adversarial attack and AutoPrompt (Shin et al., 2020) were utilized to develop the first automatic jailbreak (Zou et al., 2023). This was achieved by projecting gradients over a vocabulary to select optimal token substitutions for a suffix or by simply adding left-side noise to the prompt (Liu et al., 2024c). Similarly, confidence elicitation could be an interesting concept to enhance chatbot jailbreaks. For instance, it might enable prompt-level multi-turn dialog attacks, such as PAIR (Chao et al., 2023) and Tree of Attacks (Mehrotra et al., 2024), to query the probability of the model's answers during a multi-turn dialog and use this information as feedback. A similar approach could be explored for adversarial misalignment (Carlini et al., 2023).

## C  PROMPTS

Table 6 shows the prompt we use to first perform a prediction on $x$=‘text’ and then elicit confidence. This prompt is a combination of methods from (Lin et al., 2022), where verbal confidence is utilized, and (Tian et al., 2023), where a two-shot approach is used for confidence elicitation.

## D  FURTHER IMPLEMENTATION DETAILS

We use 12 Nvidia A40 GPUs for our testing, every test can be conducted on only 1 A40GPU. For our tests we perturb 500 samples on 1 A40 GPU with 46GB of memory.

| Verbal Elicitation Verb. 2S k guesses prompt example | |
|---|---|
| **Prediction Prompt** | f"""{self.start_prompt_header}
Provide your k best guess for the following text (positive, negative).
Give ONLY the guesses, no other words or explanation.
For example:
Guesses: (most likely guesses, either positive or negative; not a complete sentence, just the guesses!
Separated by a comma, for example [Negative, Positive, Positive, Negative ... xk])
The text is:${text}
Guesses:
{self.end_prompt_footer}""" |
| **Confidence Elicitation Prompt** | f"""{self.start_prompt_header}
You're a model that needs to give the confidence of answers being correct.
The previous prompt was:
Provide your k best guesses for the following text (positive, negative).
Give ONLY the guesses, no other words or explanation.
For example:
Guesses: (most likely guess, either positive or negative; not a complete sentence, just the guesses!)
The text is:{text} the guesses were: {guesses_output},
given these guesses provide the verbal confidences that your guesses are correct.
Give ONLY the verbal confidences, no other words or explanation.
For example:
Confidences: (the confidences, from either (Highest, High, Medium, Low, Lowest) that your guesses are correct,
without any extra commentary whatsoever, for example [Highest, High, Medium, Low, Lowest ...];
just the confidence! Separated by a coma
Confidences:
{self.end_prompt_footer}""" |

Table 6: An example of the Verb. 2S top-1 prompting technique is as follows: The first prompt generates an answer, $\hat{y}$, through the model $f_\theta$. This answer is then passed to the Confidence Elicitation Prompt as the variable 'guess_result'. Next, the second prompt is passed through $f_\theta$ to generate the verbal confidence, $\mathbf{p}_\mathcal{C}$. The variable 'text' is the sample under analysis, while the 'start_prompt_header' and 'end_prompt_footer' are model's formatting tokens

# E EXPERIMENTAL SETUP DETAILS

## E.1 DATASETS, TASKS AND MODELS

We conducted our confidence elicitation attacks on Meta-Llama-3-8B-Instruct (Touvron et al., 2023) and Mistral-7B-Instruct-v0.2 (Jiang et al., 2023) while performing classification on two common datasets to evaluate adversarial robustness: *SST-2*, *AG-News* and one modern dataset: *StrategyQA* (Geva et al., 2021).

## E.2 EVALUATION METRICS

We utilize the evaluation framework previously proposed in (Morris et al., 2020), where an evaluation set is perturbed, and we record the following data from the Total Attacked Samples ($TAS$) set: Number of Successful Attacks ($N_{succ-atk}$), Number of Failed Attacks ($N_{fail-atk}$), and Number of Skipped Attacks ($N_{skp-atk}$). We utilize these values to record the following metrics. *Clean accuracy/Base accuracy/Original accuracy*, which offers a measure of the model's performance during normal inference. *After attack accuracy/Accuracy under attack* ($A_{aft-atk} = \frac{N_{fail-atk}}{TAS}$) or (AUA), is critical, representing how effectively the attacker deceives the model across the dataset. Similarly, the *After success rate* ($A_{succ-rte} = \frac{N_{succ-atk}}{TAS-N_{skp-atk}}$) or (ASR) excludes previously misclassified samples. The paper also considers the *Semantic similarity/SemSim*, an automatic similarity index, as modeled by $d_\epsilon$ (Cer et al., 2018). We compare the original perplexity with the new perturbed sample's perplexity, calculated using a GPT-2 model. A higher perplexity indicates that the example is less natural and fluent to the language model. *Queries* denotes the number of model calls for inference. We subdivide this metric into two categories: *All Att Queries Avg* and *Succ Att Queries Avg*. The latter records the queries for successful attacks only, while the former includes all queries. Additionally, we track the duration of the attack process to perturb all the samples under *Total Attack Time*.

## E.3 EVALUATION BASELINES

We compare our guided word substitution attacks, CEAttack to "Self-Fool Word Sub" from (Xu et al., 2024), SSPAttack from (Liu et al., 2023) and TextHoaxer from (Ye et al., 2022). The "Self-Fool Word Sub" method operates by instructing the LLM model to substitute words with synonyms while

maintaining semantic integrity. We execute a query to generate $x_{adv}$, extract $x_{adv}$ from the generated string, and perform another call to the model to achieve the misclassification, if no misclassification is found, we repeat the process twenty times. On the other hand, the SSPAttack algorithm initially heavily perturbs the original sample with multiple synonym word substitutions to induce misclassification. Subsequently, they optimize the sample for quality by first reverting as many perturbed words to their original form as possible, and then by substituting words with synonyms that enhance semantic similarity. This optimization process is conducted using the hard-label as feedback. In a manner similar to SSPAttack, TextHoaxer initially introduces significant perturbations to the input $x$ to formulate an adversarial candidate. It then utilizes resources like Counter-Fitted word embeddings (Mrkšić et al., 2016) to extract the word embeddings of both the original $x$ and the adversarial version $x_{adv}$. From these embeddings, a perturbation matrix is constructed. TextHoaxer constructs a loss function that is optimized over this perturbation matrix. The optimization aims to enhance semantic similarity while adhering to two constraints: a pairwise perturbation constraint to maintain semantic closeness of word substitutions, and a sparsity constraint to control the extent of word replacements, ensuring minimal yet effective perturbations.

### E.4 Expected Calibration Error (ECE)

The ECE is calculated using the formula:

$$\text{ECE} = \sum_{m=1}^{M} \frac{|B_m|}{n} \cdot |\text{acc}(B_m) - \text{conf}(B_m)|$$

In this formula, $n$ represents the total number of samples, and $M$ is the total number of bins used to partition the predicted confidence scores. The term $B_m$ denotes the set of indices of samples whose predicted confidence falls into the $m$-th bin, and $|B_m|$ is the number of samples in this bin. The accuracy within each bin, $\text{acc}(B_m)$, is calculated as the proportion of correctly predicted samples, given by the equation $\text{acc}(B_m) = \frac{1}{|B_m|} \sum_{i \in B_m} \mathbf{1}(\hat{y}_i = y_i)$, where $\hat{y}_i$ is the predicted class label and $y_i$ is the true class label for sample $i$. The confidence of the predictions in the $m$-th bin, $\text{conf}(B_m)$, is the average of the predicted confidence scores for the samples in the bin, calculated as $\text{conf}(B_m) = \frac{1}{|B_m|} \sum_{i \in B_m} \hat{p}_i$, where $\hat{p}_i$ is the predicted probability for the predicted class of sample $i$. The ECE thus captures the weighted average of the absolute differences between accuracy and confidence across all bins, providing a summary measure of model calibration.

We use 10 bins to generate our plots in Figure 3

## F  Further Calibration Studies

### F.1  Confidence elicitation generalization

We investigate whether confidence elicitation serves as a reliable measurement of uncertainty across various models (Table 7) and datasets (Table 8). Our findings suggest that confidence elicitation is indeed a dependable tool for approximating confidence across different models.

### F.2  Self-Consistency calibration

It is possible to use empirical self-consistency (Wang et al., 2023), with the parameters set to $k = 1$, $M = 20$, and $\tau = 1$, instead of employing confidence elicitation for our attacks. This approach generates multiple predictions from the model, which we can then leverage to obtain empirical uncertainty estimates. We find that the results are similar to those achieved using confidence elicitation, as shown in Table 9. However, approximating uncertainty using this technique renders the attacks impractical, since each input perturbation would require $M$ calls to the model to estimate confidence, whereas confidence elicitation requires only a single call. We find results similar to previous work, where the outcomes are mixed. Specifically, in line with the findings of (Xiong et al., 2024), we observe that the confidence elicitation technique outperforms self-consistency on StrategyQA for uncertainty estimation.

| Calibration of verbal confidence elicitation on more models | | | | | |
|---|---|---|---|---|---|
| Model | Dataset | Avg ECE ↓ | AUROC ↑ | AUPRC Positive ↑ | AUPRC Negative ↑ |
| Gemma2 9B-Instruct | SST2 | 0.0591 | 0.9486 | 0.9547 | 0.9357 |
| | AG-News | 0.1666 | 0.8342 | - | - |
| | StrategyQA | 0.2295 | 0.6631 | 0.5899 | 0.7563 |
| Mistral-Nemo 12B-Instruct-2407 | SST2 | 0.0645 | 0.9958 | 0.9944 | 0.9970 |
| | AG-News | 0.0673 | 0.9194 | - | - |
| | StrategyQA | 0.2748 | 0.6214 | 0.6425 | 0.5863 |
| Qwen2.5 7B-Instruct | SST2 | 0.0382 | 0.9534 | 0.9399 | 0.9480 |
| | AG-News | 0.0753 | 0.8722 | - | - |
| | StrategyQA | 0.2332 | 0.6247 | 0.6649 | 0.5624 |
| LLaMa-3.2-11B Vision-Instruct | SST2 | 0.0581 | 0.9535 | 0.9645 | 0.9270 |
| | AG-News | 0.1090 | 0.8954 | - | - |
| | StrategyQA | 0.2720 | 0.6366 | 0.6532 | 0.5928 |

Table 7: Calibration of other models on core datasets SST2, AG-News and StrategyQA

| Calibration of verbal confidence elicitation on more datasets | | | | | |
|---|---|---|---|---|---|
| Model | Dataset | Avg ECE ↓ | AUROC ↑ | AUPRC Positive ↑ | AUPRC Negative ↑ |
| LLaMa-3-8B Instruct | RTE | 0.2598 | 0.8230 | 0.7972 | 0.8418 |
| | QNLI | 0.1352 | 0.8413 | 0.8561 | 0.8182 |
| Mistral-7B Instruct-v0.3 | RTE | 0.3047 | 0.6507 | 0.6032 | 0.6927 |
| | QNLI | 0.2764 | 0.6951 | 0.6444 | 0.7345 |

Table 8: Calibration of other datasets on core models Mistral and LLaMa3

## F.3 MORE CALIBRATION PLOTS

We show the reliability plots for mistral in Figure 6

## G EVALUATION ON A CLOSE-SOURCE API MODEL

We conducted tests on GPT-4o using the OpenAI API. We found this model to be more robust against word substitutions (Table 11) and better at eliciting confidence (Table 10).

## H ABLATION

Since our work introduces the concept of confidence elicitation attacks, we follow previous adversarial attack work and set the temperature $\tau$ to approximately 0. This approach maintains the general purposefulness of the model while allowing deterministic behavior across multiple model calls with the same input $M$ in Figure 1. Setting $\tau \approx 0$ also has the added benefit of reducing computation costs for our analysis since we won't have to perform multiple calls to the model after each perturbation. Nonetheless, we provide an ablation study with $\tau = 0.7$ in Appendix H.3.

### H.1 ABLATION ON NUMBER OF EMBEDDING $|S|$

The following table 12 holds the values in figure 4 for the ablation on $|S|$ plots.

### H.2 ABLATION ON MAXIMUM NUMBER OF WORD SUBSTITUTIONS $|W|$

The following table 13 holds the values in figure 4 for the ablation on $|W|$ plots.

### H.3 TEMPERATURE ABLATION

We conduct the same experiments in Section 5.2 with a temperature of 0.7, the findings are shown in Tables 14, 15, 16.

| Calibration of empirical self-consistency | | | | | | |
|---|---|---|---|---|---|---|
| Model | Dataset | Uncertainty Estimation Technique | Avg ECE ↓ | AUROC ↑ | AUPRC Positive ↑ | AUPRC Negative ↑ |
| LLaMa-3-8B Instruct | SST2 | Self-Consistency | 0.0515 | 0.9631 | 0.9730 | 0.9433 |
| | | Confidence Elicitation | 0.1264 | 0.9696 | 0.9730 | 0.9678 |
| | AG-News | Self-Consistency | 0.0774 | 0.9147 | - | - |
| | | Confidence Elicitation | 0.1376 | 0.9293 | - | - |
| | StrategyQA | Self-Consistency | 0.2113 | 0.6975 | 0.6639 | 0.7124 |
| | | Confidence Elicitation | 0.0492 | 0.6607 | 0.6212 | 0.6863 |
| Mistral-7B Instruct-v0.3 | SST2 | Self-Consistency | 0.0675 | 0.9466 | 0.9418 | 0.9255 |
| | | Confidence Elicitation | 0.1542 | 0.9537 | 0.9616 | 0.9343 |
| | AG-News | Self-Consistency | 0.0837 | 0.9240 | - | - |
| | | Confidence Elicitation | 0.1216 | 0.8826 | - | - |
| | StrategyQA | Self-Consistency | 0.3671 | 0.6182 | 0.6416 | 0.5861 |
| | | Confidence Elicitation | 0.1295 | 0.6358 | 0.6421 | 0.6185 |

Table 9: Calibration of empirical self-consistency

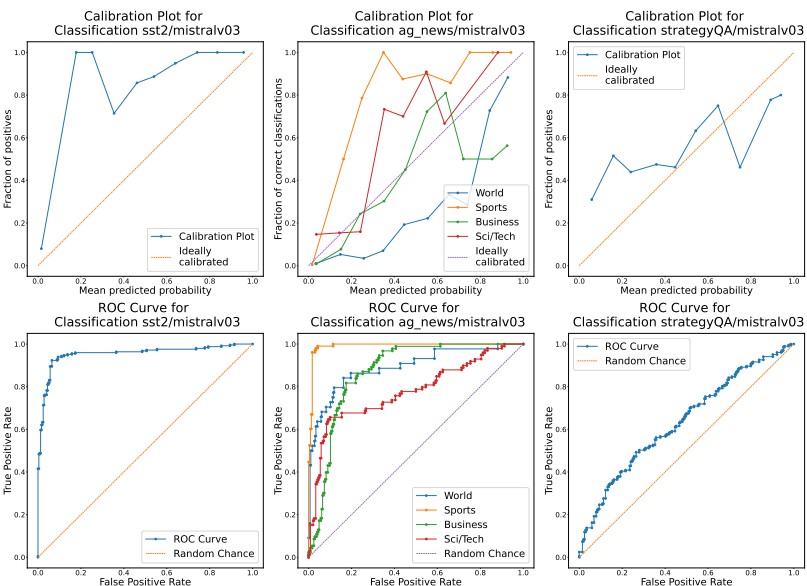

Figure 6: Reliability plots. On the top, we show the SST2, AG-News and StrategyQA on Mistralv0.3 7B Instruct calibration plots. On the bottom, the ROC curves. The optimal calibration corresponds to a diagonal line.

## H.4 DELETE WORD RANKING SCHEMA ABLATION

It is possible to enhance the efficacy of the attack by initially ranking the input words based on their importance using a word deletion ranking schema (Table 17). This involves removing each word from the input example, one at a time, and observing the change in confidence elicited in the output. Words that cause the largest change in confidence are ranked higher, while those causing minimal change are ranked lower. Once the words are ranked, we proceed to perform CEAttacks as previously done. A word deletion ranking schema is appropriate for our technique because it adheres to the black box constraints of the attack. Alternative ranking methods, such as using attention scores or word saliency, would require some knowledge of the model's inner workings.

## H.5 SIMPLE CONFIDENCE ELICITATION ATTACKS

We can replace the Dirichlet aggregator by setting $k = 1$, and instead of using verbal confidence (VC), we employ numerical verbal confidence (NVC). In this approach, we ask the model to provide its confidence numerically as a value between 0 and 1 for a prediction. We find that the performance

| Calibration of verbal confidence elicitation on an API model | | | | | |
|---|---|---|---|---|---|
| Model | Dataset | Avg ECE ↓ | AUROC ↑ | AUPRC Positive ↑ | AUPRC Negative ↑ |
| GPT-4o 2024-08-06 | SST2 | 0.0286 | 0.9713 | 0.0297 | 0.0274 |
| | AG-News | 0.0641 | 0.9306 | - | - |
| | StrategyQA | 0.2300 | 0.7410 | 0.2373 | 0.2227 |

Table 10: Calibration of GPT-4o

| | | | | | Attack performance on an API model | | | |
|---|---|---|---|---|---|---|---|---|
| Model | Dataset | Technique | CA [%] ↑ | AUA [%] ↓ | ASR [%] ↑ | SemSim ↑ | Succ Att Queries Avg ↓ | Total Attack Time [HHH:MM:SS] ↓ |
| GPT-4o 2024-08-06 | SST2 | SSPAttack | 94.0 | 89.0 | 5.32 | 0.86 | 60.8 | 001:58:12 |
| | | CEAttack | 96.0 | 82.0 | 14.58 | 0.88 | 41.92 | 002:54:00 |
| | | CEAttack++ | 95.0 | 68.0 | 28.42 | 0.87 | 108.33 | 005:34:25 |
| | AG-News | SSPAttack | 88.0 | 87.0 | 1.14 | 0.87 | 144.0 | 002:37:35 |
| | | CEAttack | 87.0 | 79.0 | 9.2 | 0.92 | 82.0 | 005:33:36 |
| | | CEAttack++ | 88.0 | 75.0 | 14.77 | 0.91 | 412.23 | 025:56:00 |
| | StrategyQA | SSPAttack | 65.0 | 52.0 | 20.0 | 0.9 | 19.07 | 000:13:22 |
| | | CEAttack | 64.0 | 45.0 | 29.69 | 0.89 | 21.15 | 000:31:01 |
| | | CEAttack++ | 68.0 | 43.0 | 36.76 | 0.88 | 39.52 | 001:20:35 |

Table 11: Confidence elicitation attacks can also target closed-source API models. Naturally, their larger scale makes them more robust to semantic perturbations. Therefore, we set |S| to 20 for SSPAttack and CEAttack. For CEAttack++, we set |S| to 50 across all datasets, employ a delete-word ranking scheme, and specify |W| as 5 for StrategyQA, 10 for SST2, and 20 for AG-News. These configurations represent the best set of hyperparameters identified through our ablation studies.

of the attack is lower (Table 18), likely due to having a weaker feedback signal with less fine-grained thresholds.

# I  QUALITATIVE EXAMPLES

Word substitutions using Counter-fitted embeddings are already constrained in terms of semantics due to the pre-built nature of the dictionary. This ensures that each word can only be replaced with a previously vetted synonym. Compared to "Self-Fool Word Sub" and SSPAttack our method more effectively preserves the original meaning of the text, as evidenced by the high "SemSim" in Table 3. Table 19 provides multiple examples from SST2 that have been perturbed by our proposed technique, CEAttack demonstrating conversions of examples from Positive to Negative and from Negative to Positive sentiment. Additional examples for AG-News and StrategyQA can be found in Table 20 and Table 21 in the 'More qualitative examples' Section J.

# J  MORE QUALITATIVE EXAMPLES

## J.1  AG NEWS

## J.2  STRATEGY QA

| | | | | | | | Ablation on $\lvert S \rvert$ CEAttack | | | | |
|---|---|---|---|---|---|---|---|---|---|---|---|
| Model | Dataset | $\lvert S \rvert$ | CA [%]↑ | AUA [%]↓ | ASR [%]↑ | SemSim↑ | Original Perplexity↓ | After-Attack Perplexity↓ | All Att Queries Avg↓ | Succ Att Queries Avg↓ | Total Attack Time [HHH:MM:SS]↓ |
| LLaMa-3-8B Instruct | SST2 | 1 | 85.86 | 78.79 | 8.24 | 0.9 | 61.74 | 105.6 | 4.53 | 4.71 | 000:44:32 |
| | | 5 | 87.88 | 69.7 | 20.69 | 0.89 | 81.73 | 131.95 | 13.45 | 16.11 | 002:11:18 |
| | | 10 | 85.86 | 68.69 | 20.0 | 0.89 | 75.06 | 105.2 | 21.81 | 28.52 | 003:22:45 |
| | | 20 | 85.86 | 63.64 | 25.88 | 0.89 | 72.24 | 106.77 | 34.56 | 49.36 | 005:19:58 |
| | | 50 | 86.87 | 58.59 | 32.56 | 0.88 | 74.28 | 112.37 | 61.95 | 78.85 | 009:30:16 |
| | AG-News | 1 | 69.0 | 62.0 | 10.14 | 0.92 | 132.98 | 143.96 | 5.71 | 5.71 | 000:48:54 |
| | | 5 | 68.0 | 57.0 | 16.18 | 0.93 | 73.76 | 92.56 | 23.85 | 23.45 | 003:02:26 |
| | | 10 | 69.0 | 57.0 | 17.39 | 0.91 | 93.73 | 122.97 | 43.61 | 38.5 | 005:47:23 |
| | | 20 | 69.0 | 52.0 | 24.64 | 0.92 | 67.26 | 88.92 | 75.32 | 75.35 | 009:41:50 |
| | | 50 | 68.0 | 47.0 | 30.88 | 0.91 | 63.93 | 95.39 | 132.49 | 130.19 | 016:30:55 |
| | StrategyQA | 1 | 63.0 | 55.0 | 12.7 | 0.9 | 123.55 | 201.62 | 1.95 | 2.75 | 000:05:25 |
| | | 5 | 64.0 | 39.0 | 39.06 | 0.9 | 109.45 | 185.19 | 5.38 | 6.68 | 000:12:03 |
| | | 10 | 65.0 | 31.0 | 52.31 | 0.89 | 109.8 | 216.1 | 9.14 | 11.35 | 000:20:29 |
| | | 20 | 65.0 | 26.0 | 60.0 | 0.89 | 99.77 | 184.98 | 14.98 | 18.66 | 000:32:50 |
| | | 50 | 63.0 | 24.0 | 61.9 | 0.89 | 113.4 | 202.82 | 28.24 | 35.74 | 000:58:51 |

Table 12: How the greedy search process is affected if we increase the number of potential synonyms per word $\lvert S \rvert$.

| | | | | | | | Ablation on $\lvert W \rvert$ CEAttack | | | | |
|---|---|---|---|---|---|---|---|---|---|---|---|
| Model | Dataset | $\lvert W \rvert$ | CA [%]↑ | AUA [%]↓ | ASR [%]↑ | SemSim↑ | Original Perplexity↓ | After-Attack Perplexity↓ | All Att Queries Avg↓ | Succ Att Queries Avg↓ | Total Attack Time [HHH:MM:SS]↓ |
| LLaMa-3-8B Instruct | SST2 | 1 | 85.86 | 75.76 | 11.76 | 0.89 | 84.46 | 115.06 | 11.46 | 12.4 | 001:49:00 |
| | | 5 | 85.86 | 68.69 | 20.0 | 0.89 | 75.06 | 105.2 | 21.81 | 28.52 | 003:23:17 |
| | | 10 | 87.88 | 63.64 | 27.59 | 0.87 | 68.2 | 111.02 | 26.33 | 31.70 | 004:14:46 |
| | | 15 | 85.86 | 65.66 | 23.53 | 0.89 | 72.11 | 108.58 | 28.39 | 37.55 | 004:22:39 |
| | | 20 | 85.86 | 66.67 | 22.35 | 0.87 | 62.32 | 101.31 | 27.85 | 36.73 | 004:12:58 |
| | AG-News | 1 | 67.0 | 61.0 | 8.96 | 0.93 | 113.8 | 132.98 | 15.57 | 20.0 | 001:59:58 |
| | | 5 | 69.0 | 57.0 | 17.39 | 0.91 | 93.73 | 122.97 | 43.61 | 38.5 | 005:47:23 |
| | | 10 | 69.0 | 50.0 | 27.54 | 0.91 | 78.46 | 115.21 | 78.35 | 72.73 | 010:15:10 |
| | | 15 | 69.0 | 49.0 | 28.99 | 0.91 | 68.93 | 114.29 | 106.51 | 99.5 | 014:17:19 |
| | | 20 | 71.0 | 47.0 | 33.8 | 0.91 | 67.15 | 116.75 | 133.87 | 113.58 | 018:35:28 |
| | StrategyQA | 1 | 63.0 | 48.0 | 23.81 | 0.89 | 107.11 | 226.42 | 4.08 | 8.53 | 000:09:21 |
| | | 5 | 65.0 | 31.0 | 52.31 | 0.89 | 109.8 | 216.1 | 9.14 | 11.35 | 000:20:29 |
| | | 10 | 64.0 | 32.0 | 50.0 | 0.89 | 105.48 | 209.16 | 9.42 | 11.78 | 000:20:49 |
| | | 15 | 64.0 | 32.0 | 50.0 | 0.89 | 105.48 | 209.16 | 9.42 | 11.78 | 000:20:49 |
| | | 20 | 64.0 | 32.0 | 50.0 | 0.89 | 105.48 | 209.16 | 9.42 | 11.78 | 000:20:49 |

Table 13: How the greedy search process is affected if we increase the maximum number of potential word substitutions in the sentence $\lvert W \rvert$.

| | | Attack Performance Tests | | | | | | | | |
|---|---|---|---|---|---|---|---|---|---|---|
| | | Self-Fool Word Sub | | | SSPAttack | | | CEAttack | | |
| Model | Dataset | CA [%]↑ | AUA [%]↓ | ASR [%]↑ | CA [%]↑ | AUA [%]↓ | ASR [%]↑ | CA [%]↑ | AUA [%]↓ | ASR [%]↑ |
| LLaMa-3-8B Instruct | SST2 | 89.9 | 87.47 | 2.7 | 89.31 | 84.48 | 5.42 | 90.12 | **57.86** | **35.79** |
| | AG-News | - | **-** | **-** | 61.96 | 50.1 | 19.14 | 59.8 | **35.96** | **39.86** |
| | StrategyQA | 61.12 | 59.32 | 2.95 | 63.53 | 37.27 | 41.32 | 62.12 | **23.05** | **62.9** |
| Mistral-7B Instruct-v0.3 | SST2 | 89.77 | 86.01 | 4.19 | 89.6 | 80.67 | 9.98 | 89.38 | **66.46** | **25.64** |
| | AG-News | 63.82 | 62.65 | 1.84 | 65.14 | 58.1 | 10.8 | 65.86 | **14.2** | **78.44** |
| | StrategyQA | 58.68 | 58.44 | 0.42 | 54.9 | 34.31 | 37.5 | 55.88 | **23.26** | **58.37** |

Table 14: Results of performing Confidence Elicitation Attacks when the model has a temperature of 0.7. Numbers in **bold** represent the best results

| | | Quality Tests | | | | | | | | |
|---|---|---|---|---|---|---|---|---|---|---|
| | | Self-Fool Word Sub | | | SSPAttack | | | CEAttack | | |
| Model | Dataset | SemSim↑ | Original Perplexity↓ | After-Attack Perplexity↓ | SemSim↑ | Original Perplexity↓ | After-Attack Perplexity↓ | SemSim↑ | Original Perplexity↓ | After-Attack Perplexity↓ |
| LLaMa-3-8B Instruct | SST2 | 0.87 | 80.94 | 99.46 | 0.89 | 76.8 | 148.38 | **0.88** | 66.49 | 105.88 |
| | AG-News | - | - | - | 0.88 | 74.61 | 208.4 | **0.93** | 66.82 | 84.68 |
| | StrategyQA | 0.88 | 112.97 | 131.68 | 0.91 | 110.59 | 210.26 | **0.89** | 99.98 | 178.08 |
| Mistral-7B Instruct-v0.3 | SST2 | 0.87 | 71.05 | 82.83 | 0.89 | 64.51 | 113.8 | **0.88** | 66.14 | 96.25 |
| | AG-News | 0.86 | 73.73 | 71.49 | 0.87 | 71.43 | 171.11 | **0.94** | 59.48 | 72.65 |
| | StrategyQA | **0.93** | 120.92 | 169.04 | 0.92 | 90.23 | 190.47 | 0.9 | 98.48 | 180.53 |

Table 15: Quality results of performing Confidence Elicitation Attacks when the model has a temperature of 0.7. Numbers in **bold** represent the best results for semantic similarity, only successful perturbations are considered.

| | | Efficiency Test | | | | | | | | |
| | | Self-Fool Word Sub | | | SSPAttack | | | CEAttack | | |
| Model | Dataset | All Att Queries Avg ↓ | Succ Att Queries Avg ↓ | Total Attack Time [HHH:MM:SS] ↓ | All Att Queries Avg ↓ | Succ Att Queries Avg ↓ | Total Attack Time [HHH:MM:SS] ↓ | All Att Queries Avg ↓ | Succ Att Queries Avg ↓ | Total Attack Time [HHH:MM:SS] ↓ |
|---|---|---|---|---|---|---|---|---|---|---|
| LLaMa-3-8B Instruct | SST2 | 20.92 | 3.0 | 001:45:13 | 7.24 | 70.04 | 043:17:07 | 23.48 | 27.28 | 020:35:45 |
| | AG-News | - | - | - | 34.66 | 165.36 | 073:29:13 | 43.17 | 42.30 | 028:16:30 |
| | StrategyQA | 21.78 | 3.0 | 000:41:37 | 10.79 | 21.47 | 002:16:44 | 8.45 | 10.54 | 001:27:07 |
| Mistral-7B Instruct-v0.3 | SST2 | 20.43 | 3.0 | 001:18:53 | 10.4 | 72.04 | 038:18:17 | 23.74 | 25.06 | 018:02:37 |
| | AG-News | 20.85 | 3.0 | 001:20:40 | 20.59 | 157.26 | 062:05:10 | 43.86 | 44.68 | 018:02:17 |
| | StrategyQA | 21.01 | 3.0 | 000:34:38 | 9.18 | 19.33 | 001:35:31 | 8.71 | 10.90 | 001:40:31 |

Table 16: Efficiency results of performing Confidence Elicitation Attacks when the model has a temperature of 0.7.

| | | Attack Performance with a delete word ranking schema for CEAttacks | | | | | | | | | | |
| | | CA [%] ↑ | | AUA [%] ↓ | | ASR [%] ↑ | | SemSim ↑ | | Succ Att Queries Avg ↓ | | Total Attack Time [HHH:MM:SS] ↓ | |
| Model | Dataset | Random Ranking | Delete Ranking | Random Ranking | Delete Ranking | Random Ranking | Delete Ranking | Random Ranking | Delete Ranking | Random Ranking | Delete Ranking | Random Ranking | Delete Ranking |
|---|---|---|---|---|---|---|---|---|---|---|---|---|---|
| LLaMa-3-8B Instruct | SST2 | 90.56 | 90.76 | 72.69 | 65.06 | 19.73 | 28.32 | 0.88 | 0.88 | 25.60 | 35.67 | 017:30:57 | 025:33:37 |
| | AG-News | 62.17 | 61.97 | 43.06 | 40.85 | 30.74 | 34.09 | 0.93 | 0.93 | 42.36 | 68.13 | 024:31:58 | 039:22:19 |
| | StrategyQA | 60.12 | 59.92 | 32.67 | 33.07 | 45.67 | 44.82 | 0.89 | 0.89 | 10.95 | 17.10 | 001:25:34 | 002:19:30 |
| Mistral-7B Instruct-v0.3 | SST2 | 87.45 | 88.08 | 71.76 | 67.78 | 17.94 | 23.04 | 0.88 | 0.88 | 24.54 | 33.18 | 017:13:44 | 024:05:07 |
| | AG-News | 66.18 | 65.89 | 40.82 | 33.24 | 38.33 | 49.56 | 0.93 | 0.92 | 42.66 | 68.95 | 017:16:52 | 027:49:15 |
| | StrategyQA | 59.61 | 58.87 | 36.21 | 33.0 | 39.26 | 43.93 | 0.9 | 0.89 | 11.37 | 18.01 | 001:43:48 | 002:43:34 |

Table 17: Confidence elicitation can also serve as a proxy for ranking the importance of words in the input.

| | | Confidence Elicitation Attack with a simple confidence elicitation technique | | | | | |
| Model | Dataset | | CA [%] ↑ | AUA [%] ↓ | ASR [%] ↑ | SemSim ↑ | Succ Att Queries Avg ↓ | Total Attack Time [HHH:MM:SS] ↓ |
|---|---|---|---|---|---|---|---|---|
| LLaMa-3-8B Instruct | SST2 | NVC | 91.2 | 76.0 | 16.67 | 0.89 | 29.5 | 005:11:15 |
| | | Dirichlet+VC | 90.56 | 72.69 | 19.73 | 0.88 | 25.60 | 017:30:57 |
| | StrategyQA | NVC | 65.0 | 48.2 | 25.85 | 0.89 | 12.22 | 000:52:20 |
| | | Dirichlet+VC | 60.12 | 32.67 | 45.67 | 0.89 | 10.95 | 001:25:34 |

Table 18: Confidence Elicitation Attack with a simple confidence elicitation technique

| Qualitative Example SST2 (Sentiment Classification) | | | | | | | | |
| Technique | Sample | SemSim | Perplexity | Perturbed Words | Ground Truth | Prediction | Empirical Mean (Score) |
|---|---|---|---|---|---|---|---|
| Original | there is nothing outstanding about this film, but it is good enough and will likely be appreciated most by sailors and folks who know their way around a submarine. | - | 48.59 | - | Positive | Positive | 0.48 |
| CEAttack (Ours) | there is nothing outstanding about this film, but it is **appropriate** enough and will likely be appreciated most by sailors and **males** who know their **routes** around a submarine. | 0.85 | 114.7 | 3 | Positive | Negative | 0.90 |
| Original | the movie achieves as great an impact by keeping these thoughts hidden as ... (quills) did by showing them. | - | 407.19 | - | Positive | Positive | 0.40 |
| CEAttack (Ours) | the **filmmakers obtains** as **formidable** an impact by keeping these thoughts hidden as ... (**plume**) did by showing them. | 0.84 | 616.55 | 4 | Positive | Negative | 0.50 |
| Original | combining quick-cut editing and a blaring heavy metal much of the time, beck seems to be under the illusion that he's shooting the latest system of a down video. | - | 328.26 | - | Negative | Negative | 0.26 |
| CEAttack (Ours) | **mixing** quick-cut editing and a **thundering** heavy metal much of the **periods**, beck **appear** to **get** under the **trickery** that he's shooting the latest system of a down video. | 0.84 | 571.93 | 6 | Negative | Positive | 0.75 |
| Original | schaeffer has to find some hook on which to hang his persistently useless movies, and it might as well be the resuscitation of the middle-aged character. | - | 95.63 | - | Negative | Negative | 0.39 |
| CEAttack (Ours) | **colson** has to find some hook on which to hang his persistently **incongruous film**, and it **may** as **alright get** the resuscitation of the middle-aged character. | 0.84 | 157.87 | 6 | Negative | Positive | 0.65 |

Table 19: Confidence elicitation attacks and their confidence levels. Perturbed words are in **bold**.

**Qualitative Example**
**AG-News (News Classification) LLaMa-3-8B-Instruct**

| Technique | Sample | SemSim | Perplexity | Perturbed Words | Ground Truth | Prediction | Empirical Mean (Score) |
|---|---|---|---|---|---|---|---|
| Original | Producer sues for Rings profits Hollywood producer Saul Zaentz sues the producers of The Lord of the Rings for $20m in royalties., | - | 257.14 | - | World | World | 0.50 |
| CEAttack (Ours) | **Producing** sues for Rings profits Hollywood producer Saul Zaentz sues the producers of The Lord of the Rings for $20m in royalties. | 0.94 | 325.36 | 1 | World | Business | 0.66 |
| Original | Injured Heskey to miss England friendly NEWCASTLE, England (AP) - Striker Emile Heskey has pulled out of the England squad ahead of Wednesday #39;s friendly against Ukraine because of a tight hamstring, the Football Association said Tuesday. | - | 213.64 | - | Sport | Sport | 0.46 |
| CEAttack (Ours) | **Wound** Heskey to **senorita** England friendly NEWCASTLE, **English** (AP) - Striker Emile Heskey has pulled out of the **Britannica** squad ahead of Wednesday #39;s friendly against Ukraine because of a **intensive** hamstring, the Football Association said Tuesday. | 0.87 | 354.76 | 5 | Sport | World | 0.57 |
| Original | SEC may put end to quid pro quo (USATODAY.com) USATODAY.com - The Securities and Exchange Commission is expected to vote Wednesday to prohibit mutual fund companies from funneling stock trades to brokerage firms that agree to promote their funds to investors. | - | 56.46 | - | Business | Business | 0.47 |
| CEAttack (Ours) | SEC may put **ends** to quid pro quo (USATODAY.com) USATODAY.com - The Securities and Exchange Commission is expected to **voices** Wednesday to prohibit **reciprocated** fund companies from funneling stock trades to brokerage firms that **ok** to promote their funds to investors. | 0.87 | 140.03 | 4 | Business | World | 0.6 |
| Original | IBM Seeks To Have SCO Claims Dismissed (NewsFactor) NewsFactor - IBM (NYSE: IBM) has – again – sought to have the pending legal claims by The SCO Group dismissed. According to a motion it filed in a U.S. district court, IBM argues that SCO has no evidence to support its claims that it appropriated confidential source code from Unix System V and placed it in Linux. | - | 95.57 | - | Sci/Tech | Sci/Tech | 0.61 |
| CEAttack (Ours) | IBM Seeks To Have SCO Claims Dismissed (NewsFactor) NewsFactor - IBM (NYSE: IBM) has – again – sought to have the pending legal claims by The SCO **Clusters** dismissed. According to a motion it filed in a U.S. district court, IBM argues that SCO has no evidence to support its claims that it appropriated confidential source code from Unix System **volts** and placed it in Linux. | 0.94 | 111.51 | 2 | Sci/Tech | Business | 0.63 |

Table 20: Examples of confidence elicitation attacks and their respective confidence levels: Top) A positive example perturbed to negative, Bottom) A negative example perturbed to positive. Perturbed words are in **bold**.

**Qualitative Example**
**StrategyQA (Reasoning Classification) LLaMa-3-8B-Instruct**

| Technique | Sample | SemSim | Perplexity | Perturbed Words | Ground Truth | Prediction | Empirical Mean (Score) |
|---|---|---|---|---|---|---|---|
| Original | Did the Wehrmacht affect the outcome of the War to End All Wars? | - | 33.2 | - | False | False | 0.49 |
| CEAttack (Ours) | Did the Wehrmacht **impacting** the outcome of the War to **Conclude** All Wars? | 0.89 | 112.07 | 2 | False | True | 0.62 |
| Explanation | The Wehrmacht was the unified military of Germany from 1935 to 1945 The War to End All Wars is a nickname for World War I World War I ended in 1918 | | | | | | |
| Original | Does Mercury make for good Slip N Slide material? | - | 1224.48 | - | False | False | 0.13 |
| CEAttack (Ours) | Does Mercury **deliver** for **best** Slip N Slide material? | 0.86 | 5038.86 | 2 | False | True | 0.62 |
| Explanation | The Slip N Slide was an outdoor water slide toy. Mercury is a thick liquid at room temperature. Mercury is poisonous and used to kill hatters that lined their hats with the substance. | | | | | | |
| Original | Would human race go extinct without chlorophyll? | - | 91.15 | - | True | True | 0.43 |
| CEAttack (Ours) | Would **humanistic** race **will extinct** without chlorophyll? | 0.85 | 433.3 | 3 | True | False | 0.69 |
| Explanation | Chlorophyll is a pigment in plants responsible for photosynthesis. Photosynthesis is the process by which plants release oxygen into the atmosphere. Humans need oxygen to live. | | | | | | |
| Original | Are more people today related to Genghis Khan than Julius Caesar? | - | 294.74 | - | True | True | 0.34 |
| CEAttack (Ours) | Are more people today **connected** to Genghis Khan than Julius Caesar? | 0.94 | 289.71 | 1 | True | False | 0.54 |
| Explanation | Julius Caesar had three children. Genghis Khan had sixteen children. Modern geneticists have determined that out of every 200 men today has DNA that can be traced to Genghis Khan. | | | | | | |

Table 21: Examples of confidence elicitation attacks and their respective confidence levels: Top) A positive example perturbed to negative, Bottom) A negative example perturbed to positive. Perturbed words are in **bold**.

