# OpenReview forum: "Confidence Elicitation: A New Attack Vector for Large Language Models"
_ICLR.cc/2025/Conference — ICLR 2025 Poster_

### Official Review · Reviewer_mH4K · 2024-10-17

**Soundness:** 3
**Presentation:** 3
**Contribution:** 3
**Rating:** 8
**Confidence:** 3

**Summary:**

This paper shows how estimation of LLMs' confidence can be obtained even from black box models by directly asking models for their confidence (aggregating their answers over multiple queries), and how these continuous values can be used to generate better adversarial attacks.

It uses this approach to attack Llama3-8B and Mistral-7B-v0.3 on NLP classification tasks (SST2, AG-News and StrategyQA):
* It uses these estimates to create better word-substitution attacks using a hill-climbing approach, which achieves higher ASR with smaller perturbations than baseline methods that use hard labels or no labels.
* It qualitatively analyzes these attacks and their intermediate steps.

**Strengths:**

* The idea of estimating the confidence of black-box LLMs to inform adversarial attacks is interesting.
* The presentation and the figures are clean and clear. It explains its ideas well.
* The attacks are compared to baselines and a detailed sensitivity analysis on both the main technique and baselines is performed.
* The attacks themselves are relatively interpretable, and this tool could present a nice starting point for future work aimed at understanding how LLMs perform classification tasks.

**Weaknesses:**

* The confidence estimation method presented here is never evaluated properly.
  * Its performance or calibration is never measured.
  * It is not compared to simple baselines (e.g. simply making multiple queries at temperature 1 and measuring the frequency of each answer).
* It is unclear if the attack presented is relevant for the threat models described in the introduction (jailbreaks and adversarial misalignment): because in the case of harmful requests, the probability of a harmful completion is tiny, and models might put maximum confidence in their rejection, CEAttack might not get any continuous signal, and thus might not outperform hard-label methods. Other threat models might have been relevant (e.g. attacking LLM classifiers behind an API) but these also have properties which make attacks hard (e.g. it might not be possible to ask them questions).
* The attack is never performed on real attack scenarios, or on real black-box models (which might have uncovered the issue above).
* The attacks are not very impressive: the ASR is low in absolute terms, and when looking at the attacks, many of them seem to work only because they change the meaning of the sentence or make the sentence less meaningful. This is particularly bad for StrategyQA (e.g. "Did the Wehrmacht affect the outcome of the War to End All Wars?" --> "Did the Wehrmacht impacting the outcome of the War to Conclude All Wars?")
* Black-box jailbreaking techniques like TAP already use similar ideas of more-or-less adherence to the target, and achieve higher performance in more difficult and realistic scenarios.

Minor points:
* The text is not clear about whether the experiments are conducted on Mistral v0.2 or v0.3.
* The abstract should probably say what size of model was studied (which matters at least as much as the version number).
* On line 267, the ASR is defined as the after success rate (it is the attack success rate)
* The text in most tables is too small.

**Questions:**

* How good is the performance of the calibration estimation? How does it compare to simple baselines (like multiple queries at T=1)?
* How would CEAttack perform with the complex confidence estimation replaced by a simpler confidence estimation technique?
* What is a real world scenario where you expect your technique (or a technique inspired by your technique) to be relevant?

---

> ### Author Response · Authors · 2024-11-27
> **Response to reviewer mH4K [1/7]**
>
> We thank the reviewer for the valuable feedback and comments
>
> > #### **1) The confidence estimation method presented here is never evaluated properly.**
>
> We have evaluated our method against a baseline and showed that the estimates produced are calibrated, as detailed in our responses below.
>
>
> > #### **1.1) Its performance or calibration is never measured.**
>
> Our results in Table 1 show that the elicited confidences are indeed calibrated, as demonstrated by the low Expected Calibration Errors (ECE) observed.
>
> > #### **1.2) It is not compared to simple baselines (e.g. simply making multiple queries at temperature 1 and measuring the frequency of each answer).**
>
> We have included a comparison to the suggested baseline. Specifically, we used empirical self-consistency [6], with the parameters set to $k=1$, $M=20$, and $\tau=1$, instead of employing confidence elicitation for our attacks. This approach generates multiple predictions from the model, which we can then leverage to obtain empirical uncertainty estimates. We find that the results are similar to those achieved using confidence elicitation, as shown in Table 9. However, approximating uncertainty using this technique renders the attacks impractical, since each input perturbation would require $M$ calls to the model to estimate confidence, whereas confidence elicitation requires only a single call. Our findings reflect those of previous work, where the outcomes are mixed. Specifically, in line with the findings of [3], we observe that the confidence elicitation technique outperforms self-consistency on StrategyQA for uncertainty estimation.
>
> We added the table for this experiment in the Appendix F.2 'Self-Consistency calibration' in Table 9.
>
>
> **Calibration of empirical self-consistency**
>
> | Model                      | Dataset    | Uncertainty Estimation Technique | Avg ECE ↓ | AUROC ↑ | AUPRC Positive ↑ | AUPRC Negative ↑ |
> |----------------------------|------------|----------------------------------|-----------|---------|-----------------|-----------------|
> | LLaMa-3-8B Instruct        | SST2       | Self-Consistency                 | 0.0515    | 0.9631  | 0.9730          | 0.9433          |
> |                            |            | Confidence Elicitation          | 0.1264    | 0.9696  | 0.9730          | 0.9678          |
> |                            | AG-News    | Self-Consistency                 | 0.0774    | 0.9147  | -               | -               |
> |                            |            | Confidence Elicitation          | 0.1376    | 0.9293  | -               | -               |
> |                            | StrategyQA | Self-Consistency                 | 0.2113    | 0.6975  | 0.6639          | 0.7124          |
> |                            |            | Confidence Elicitation          | 0.0492    | 0.6607  | 0.6212          | 0.6863          |
> | Mistral-7B Instruct-v0.3   | SST2       | Self-Consistency                 | 0.0675    | 0.9466  | 0.9418          | 0.9255          |
> |                            |            | Confidence Elicitation          | 0.1542    | 0.9537  | 0.9616          | 0.9343          |
> |                            | AG-News    | Self-Consistency                 | 0.0837    | 0.9240  | -               | -               |
> |                            |            | Confidence Elicitation          | 0.1216    | 0.8826  | -               | -               |
> |                            | StrategyQA | Self-Consistency                 | 0.3671    | 0.6182  | 0.6416          | 0.5861          |
> |                            |            | Confidence Elicitation          | 0.1295    | 0.6358  | 0.6421          | 0.6185          |
>
> *Table: Calibration of empirical self-consistency*

---

> ### Author Response · Authors · 2024-11-27
> **Response to reviewer mH4K [2/7]**
>
> > #### **2) It is unclear if the attack presented is relevant for the threat models described in the introduction (jailbreaks and adversarial misalignment): because in the case of harmful requests, the probability of a harmful completion is tiny, and models might put maximum confidence in their rejection, CEAttack might not get any continuous signal, and thus might not outperform hard-label methods. Other threat models might have been relevant (e.g. attacking LLM classifiers behind an API) but these also have properties which make attacks hard (e.g. it might not be possible to ask them questions).**
>
> We thank the reviewer for their feedback. The aim of our work is to serve as a proof-of-concept, highlighting a potential vulnerability in confidence elicitation, henceforth motivating the need for implementing appropriate guardrails in open-ended systems, particularly when these systems are paired with calibration results in next-token generation and free-form generation tasks. We hope that the community will explore the robustness of these tasks in greater detail in the future, considering that confidence elicitation has recently emerged as a candidate for uncertainty estimation in free-form generation [16, 17, 19].
>
>
> In the introduction, we include some case study scenarios as a proof of concept to demonstrate how confidence elicitation attacks might be employed specifically in a classification setting, as well as extending the conclusion to highlight how confidence elicitation attacks in free-form generation, jailbreaks and next-token prediction are all interesting avenues for future research.
>
>
> **Added in the introduction:**
>
> >Such vulnerabilities are particularly concerning in critical applications, such as healthcare [9], socio-technical systems and human-machine collaboration [8]. For example, in healthcare, where a medical system provides a diagnosis, an attacker might introduce input perturbations, aiming to achieve a misclassification. In clinical support systems, such misclassifications can have lethal consequences [10].
>
> >Providing confidence estimates through confidence elicitation, whether in a template or a free-form generation, has been shown to enhance the performance and utility of these systems. This is particularly important in domains where assessing the reliability of a model's responses is crucial for effective risk assessment, error mitigation, selective generation, and minimizing the effects of hallucinations [3]. As a result, we can anticipate these techniques to become more widespread. Consequently, exploring whether we can strengthen adversarial perturbations using confidence estimates is an important area of research, with the aim of designing more robust systems.
>
> **Added in the conclusion:**
>
> >Our results suggest the potential for confidence elicitation to enhance jailbreaks. For example, it might enable current multi-turn dialog jailbreaks [20,21] to query the probability of the model's answers and use this information as feedback. Another promising direction is to investigate how susceptible token-wise confidence elicitation is to input perturbations, and whether it is possible to control or influence the model's token selection process. Additionally, it is worth exploring how effective confidence elicitation attacks are on generative tasks such as free-form question answering [15], given that confidence elicitation has also proven to be a reliable and calibrated measure of uncertainty in generative tasks [16,17].

---

> ### Author Response · Authors · 2024-11-27
> **Response to reviewer mH4K [3/7]**
>
> Moreover, beyond the classification setting, here are two potential ways our approach can be extended to generation tasks:
>
> **1) Free-Form Generation:**
>
> There are several tasks, such as TriviaQA, NarrativeQA, PopQA, and SciQ, where we have a label set of correct answers for each datapoint. In these datasets, a common task is question answering, where the prompt is a question, and the model is expected to generate an answer in a free-form setting. In this setting, the model can generate a word sequence of arbitrary length, which is considered a generation task [16,13,12,14,15]. The accuracy of the model is evaluated based on its relevance to the label set using a threshold and an overlap algorithm, such as exact match, F1, BLEU, or ROUGE. While the original TriviaQA was evaluated using these rule-based algorithmic approaches, later works have employed language models for evaluation [19].
>
> Regarding confidence elicitation in a generative setting, recent work [16] used a ROUGE threshold of 0.3 between the generated answer and the label. Following this, the model is asked to provide its verbal confidence level. They demonstrated that using verbal confidence elicitation can achieve some of the best ECE uncertainty measures (see Table 3 (A4C) in [16]) for TriviaQA, a generative task.
>
> In the context of adversarial attacks, the adversarial setup is crucial. A straightforward and effective starting point could involve perturbing the input so that the generated answer no longer exceeds the 0.3 ROUGE/BLEU threshold [18]. This would indicate minimal overlap between the generated sentence and the label sentences. In a black-box setting, we would not know the threshold or that ROUGE is utilized. We would simply receive a binary output indicating whether the overlap is sufficient. Therefore, being able to query the model about its confidence in how well the generated answer addresses the question can serve as a valuable feedback signal. Reducing this confidence would suggest that the model is less certain that the answer truly addresses the question.
>
> There are other, deemed more challenging adversarial objectives, such as the Non-overlapping Attack, where the goal is to perturb the input so that no words in the newly generated output overlap with the target label. Another example is the Targeted Keywords Attack, where the input perturbation aims to introduce a specific keyword into the output [23]
>
> In the above setups, we would maintain the 'Verbal Elicitation Verb. 2S k guesses' elicitation framework and use the elicited confidence as guidance, just as before.
>
> Although we were unable to explore this question in detail or conduct experiments due to limited time, we believe that addressing confidence elicitation attacks in free-form generation tasks on datasets such as TriviaQA would be a natural next step for future work. This is particularly relevant given that recent papers have demonstrated that confidence elicitation can provide reliable uncertainty measures on this dataset [16]
>
> **2) Token-Wise:**
>
> Confidence elicitation can be applied to approximate token probabilities. The adversarial objective would be to minimize the probability of specific tokens so that they are no longer generated by the model, increasing the likelihood of other tokens being generated.
>
> For example, the objective may be to make the model generate a response that is unrelated to the question. We could introduce an input perturbation to minimize the probability of the token 'window' being generated in the output.
>
> Example:
> - User: You're a model that answers questions and provides the confidence for each token between 0 and 1, indicating the token's correctness. "How do you build a window?"
> - Model: To: 0.9, build: 0.95, a: 0.95, window: 0.95, you: 0.9, typically: 0.85, follow: 0.85, these: 0.9, steps: 0.9
>
> Application to Jailbreaks:
>
> This is a high-level concept, but with a similar setup, we can ask the model, "How to build a bomb." - Model: sorry: 0.8, I: 0.95, can't: 0.9, assist: 0.85, with: 0.9, that: 0.9, request: 0.85
>
> The goal would be to minimize the probability of the token 'sorry' appearing in the output, thereby increasing the model's likelihood to choose other tokens such as 'sure' or 'to'.

---

> ### Author Response · Authors · 2024-11-27
> **Response to reviewer mH4K [4/7]**
>
> > #### **3) The attack is never performed on real attack scenarios, or on real black-box models (which might have uncovered the issue above).**
>
> We have conducted some tests on GPT-4o using the OpenAI API. We found GPT-4o to be calibrated and that our attack outperforms our current best baseline (SSPAttack), though overall the model is more robust to attacks. We added the experiments in Appendix G 'Evaluation on close source API model' in Tables 10 and 11.
>
> **Calibration of verbal confidence elicitation on an API model**
>
> | Model                                     | Dataset    | Avg ECE ↓ |       |       |       | AUROC ↑ |       | AUPRC Positive ↑ |       | AUPRC Negative ↑ |       |       |       |
> |-------------------------------------------|------------|-----------|-------|-------|-------|--------|-------|------------------|-------|------------------|-------|-------|-------|
> | GPT-4o                                    | SST2       | 0.0286    |       |       |       | 0.9713 |       | 0.9672           |       | 0.9611           |       |       |       |
> | 2024-08-06                                | AG-News    | 0.0641    |       |       |       | 0.9306 |       | -                |       | -                |       |       |       |
> |                                           | StrategyQA | 0.2300    |       |       |       | 0.7410 |       | 0.7238           |       | 0.6962           |       |       |       |
>
> **Table Caption**:Calibration of GPT-4o
>
> **Attack performance on an API model**
>
> | Model   | Dataset     | Technique   | CA [%] ↑ | AUA [%] ↓ | ASR [%] ↑ | SemSim ↑ | Succ Att Queries Avg ↓ | Total Attack Time [HHH:MM:SS] ↓ |
> |---------|-------------|-------------|----------|-----------|-----------|----------|------------------------|---------------------------------|
> | GPT-4o  | SST2        | SSPAttack   | 94.0     | 89.0      | 5.32      | 0.86     | 60.8                   | 001:58:12                       |
> |         |             | CEAttack    | 96.0     | 82.0      | 14.58     | 0.88     | 41.92                  | 002:54:00                       |
> |         |             | CEAttack++  | 95.0     | 68.0      | 28.42     | 0.87     | 108.33                 | 005:34:25                       |
> |         | AG-News     | SSPAttack   | 88.0     | 87.0      | 1.14      | 0.87     | 144.0                  | 002:37:35                       |
> |         |             | CEAttack    | 87.0     | 79.0      | 9.2       | 0.92     | 82.0                   | 005:33:36                       |
> |         |             | CEAttack++  | 88.0     | 75.0      | 14.77     | 0.91     | 412.23                 | 025:56:00                       |
> |         | StrategyQA  | SSPAttack   | 65.0     | 52.0      | 20.0      | 0.9      | 19.07                  | 000:13:22                       |
> |         |             | CEAttack    | 64.0     | 45.0      | 29.69     | 0.89     | 21.15                  | 000:31:01                       |
> |         |             | CEAttack++  | 68.0     | 43.0      | 36.76     | 0.88     | 39.52                  | 001:20:35                       |
>
> **Table Caption**: Confidence elicitation attacks can also target closed-source API models. Naturally, their larger scale makes them more robust to semantic perturbations. Therefore, we set |S| to 20 for SSPAttack and CEAttack. For CEAttack++, we set |S| to 50 across all datasets, employ a delete-word ranking scheme, and specify |W| as 5 for StrategyQA, 10 for SST2, and 20 for AG-News. These configurations represent the best set of hyperparameters identified through our ablation studies.

---

> ### Author Response · Authors · 2024-11-27
> **Response to reviewer mH4K [5/7]**
>
> > #### **4) The attacks are not very impressive: the ASR is low in absolute terms, and when looking at the attacks, many of them seem to work only because they change the meaning of the sentence or make the sentence less meaningful. This is particularly bad for StrategyQA (e.g. "Did the Wehrmacht affect the outcome of the War to End All Wars?" --> "Did the Wehrmacht impacting the outcome of the War to Conclude All Wars?")**
>
> The main focus of our work is to highlight confidence elicitation as an attack vector. We further note that our proposed attack outperforms current state-of-the-art methods.
>
> With regards to semantic similarity, we concentrate on word-level attacks by perturbing the input example with a $\delta$ and using a semantic similarity encoder (USE) with a threshold of 0.84 as $\epsilon$. We have set $\epsilon$ to 0.84 across all datasets, although this can be adjusted to improve adversarial quality. For instance, given the high ASR for StrategyQA, an attacker might reasonably increase the threshold from 0.84 to 0.9. Alternative $\delta$s and $\epsilon$s exist that may achieve better quality perturbations. One potential option is using BERT-Attack [30], which employs word candidates from BERT's masked language modeling, allowing for substitutions that are more contextually fitting. For $\epsilon$, BERTScore [31] or one of the more recent contrastive-based similarity encoders could offer a better boundary [32]. While these are all viable techniques an attacker might use to enhance the quality of the attack, our work does not focus on thoroughly evaluating different $\delta$s and $\epsilon$s. Nonetheless, this represents an interesting potential direction for future research.

---

> ### Author Response · Authors · 2024-11-27
> **Response to reviewer mH4K [6/7]**
>
> > #### **5) Black-box jailbreaking techniques like TAP already use similar ideas of more-or-less adherence to the target, and achieve higher performance in more difficult and realistic scenarios.**
>
> We thank the reviewer for pointing out this paper. Although jailbreaks are a related topic, the setting is different from adversarial attacks. The aim of the highlighted paper is to develop a black-box jailbreak targeting model safeguards, where the objective is to perturb the input to generate a response that deviates from the standard refusal answer "I'm sorry, I cannot answer this request" and instead seeks to answer the malicious input question. It employs an LLM to generate multiple prompts and follows a tree structure to select the prompts with the highest potential to jailbreak the model, continuously improving these prompts to trick the target model into not refusing the question.
>
> We would love to benchmark this technique as it is very recent, but it varies significantly in several ways:
>
> **1.** It is a jailbreak and not an adversarial attack.
>
> **2.** It uses an LLM to generate a prompt, instead of word substitutions, thus making it closer to a prompt-level attack [20], rather than a word-level attack.
>
> **3.** It doesn't use confidence as guidance in any way.
>
> Given the above distinctions. We extended the work by including a relevant word-level black-box attack as an additional baseline from AAAI 2022 called **TextHoaxer** [26], which aligns with our adversarial setting, word-level attack, and black-box nature. Our method also outperforms this baseline with regards to ASR, AUA and 'Succ Att Queries Avg'.
>
> As a brief example, the table below compares the new baseline with our work.
>
> **Attack performance tests**
> Numbers in **bold** are the best results.
>
> | Model                        | Dataset    | CA [%] ↑         |            |              |            | AUA [%] ↓                               |        |        |               | ASR [%] ↑           |       |      |      |
> |------------------------------|------------|------------------|------------|--------------|------------|----------------------------------------|--------|--------|---------------|---------------------|-------|------|------|
> |                              |            | Vanilla          |            |              |            | Self-Fool Word Sub                      | **Text Hoaxer**  | SSP Attack | CE Attack       | Self-Fool Word Sub  | **Text Hoaxer**  | SSP Attack | CE Attack |
> |**LLaMa-3-8B Instruct**      | SST2       | 90.56±0.14       |            |              |            | 88.35                                  | 82.93  | 81.93  | **72.69**    | 2.22                | 8.43  | 9.73 | **19.73** |
> |                              | AG-News    | 61.62±0.38       |            |              |            | 61.17                                  | 49.3   | 45.27  | **43.06**    | 0.33                | 19.41 | 26.71 | **30.74** |
> |       | StrategyQA | 60.22±0.17       |            |              |            | 59.52                                  | 45.29  | 42.28  | **32.67**    | 1.66                | 24.67 | 29.67 | **45.67** |
> | **Mistral-7B Instruct-v0.3**      | SST2       | 87.87±0.39       |            |              |            | 84.73                                  | 74.27  | 75.31  | **71.76**    | 3.57                | 16.08 | 14.08 | **17.94** |
> |                              | AG-News    | 65.99±0.27       |            |              |            | -                                      | 48.69  | 52.48  | **40.82**    | -                   | 26.43 | 20.0 | **38.33** |
> |  | StrategyQA | 59.92±0.32       |            |              |            | 59.61                                  | 44.33  | 41.13  | **36.21**    | 1.22                | 26.23 | 30.99 | **39.26** |
>
> *Results of performing Confidence Elicitation Attacks. Numbers in bold are the best results.*
>
>
> **Furthermore, we believe that confidence elicitation could be an interesting concept to enhance jailbreaks.** For example, it could allow the Tree of Attacks to query the probability of the model's answers during a multi-turn dialog to jailbreak the model and use this information as feedback. Consequently, in the future work section in in Appendix B.2 we added a discussion of previous work on jailbreaks, highlighting past efforts such as GCG [25], PAIR [20], and TAP [21]. We then discussed that our techniques could be incorporated into these methods to enhance them.

---

> ### Author Response · Authors · 2024-11-27
> **Response to reviewer mH4K [7/7]**
>
> > #### **6) Minor points**
> We thank the reviewer for their feedback. We have added the version number for Mistral and the model sizes to the abstract, and we have corrected the typo. Once the paper is finalized, we will consider reformatting some of the tables for the camera-ready version.
>
> > #### **7) How good is the performance of the calibration estimation? How does it compare to simple baselines (like multiple queries at T=1)?**
>
> Please refer to question 1.2.
>
>
> > #### **8) How would CEAttack perform with the complex confidence estimation replaced by a simpler confidence estimation technique?**
>
> We can replace the Dirichlet aggregator, set \( k = 1 \), and instead of using verbal confidence (VC), we employ numerical verbal confidence (NVC). In this approach, we ask the model to provide its confidence numerically as a value between 0 and 1 for a prediction. We find that the performance of the attack is lower (Table 18 in Appendix H.5 'Simple Confidence elicitation attack'), likely due to having a weaker feedback signal with less fine-grained thresholds.
>
> **Confidence Elicitation Attack with a Simple Confidence Elicitation Technique**
>
> | Model                        | Dataset     | Method      | CA [%] ↑ | AUA [%] ↓ | ASR [%] ↑ | SemSim ↑ | Succ Att Queries Avg ↓ | Total Attack Time [HHH:MM:SS] ↓ |
> |------------------------------|-------------|-------------|----------|-----------|-----------|----------|-------------------------|----------------------------------|
> | LLaMa-3-8B Instruct          | SST2        | NVC         | 91.20    | 76.00     | 16.67     | 0.89     | 29.50                   | 005:11:15                        |
> |                              |             | Dirichlet+VC| 90.56    | 72.69     | 19.73     | 0.88     | 25.60                   | 017:30:57                        |
> |                              | StrategyQA  | NVC         | 65.00    | 48.20     | 25.85     | 0.89     | 12.22                   | 000:52:20                        |
> |                              |             | Dirichlet+VC| 60.12    | 32.67     | 45.67     | 0.89     | 10.95                   | 001:25:34                        |
>
>
>
> > #### **9) What is a real world scenario where you expect your technique (or a technique inspired by your technique) to be relevant?**
>
> As mentioned in previous questions, we offer a proof of concept for confidence elicitation attacks. We believe that exploring how confidence elicitation can be used in generation has significant potential, especially when combined with multi-turn dialog jailbreaks, such as those mentioned in PAIR [20] and TAP [21].
>
>
> **References**
>
> [3] Miao Xiong et al, Can LLMs Express Their Uncertainty? An Empirical Evaluation of Confidence Elicitation in LLMs, ICLR 2024
>
> [5] Jiuhai Chen et al. Quantifying Uncertainty in Answers from any Language Model and Enhancing their Trustworthiness, ACL 2024
>
> [6] Xuezhi Wang et al. self-consistency improves chain of thought reasoning in language models, ICLR 2023
>
> [8] Yusuf Sale et al. Second-Order Uncertainty Quantification: A Distance-Based Approach, ICML 2024
>
> [9] Thomas Savage et al. Large Language Model Uncertainty Measurement and Calibration for Medical Diagnosis and Treatment, 2024
>
> [20] Patrick Chao et al. PAIR Jailbreaking Black Box Large
> Language Models in Twenty Queries, NIPS workshop 2024
>
> [21] Anay Mehrotra et al. Tree of Attacks: Jailbreaking Black-Box LLMs Automatically, NIPS 2024
>
> [24] Chuan Guo et al. On Calibration of Modern Neural Networks, ICML 2017
>
> [26] Muchao Ye et al. TextHoaxer: Budgeted Hard-Label Adversarial Attacks on Text. AAAI 2022
>
> [30] Linyang Li et al. BERT-ATTACK: Adversarial Attack Against BERT Using BERT, EMNLP 2020
>
> [31] Tianyi Zhang et al. BERTScore: Evaluating Text Generation with BERT, ICLR 2020
>
> [32] Jia, Shuyue, Awesome Semantic Textual Similarity, https://github.com/SuperBruceJia/Awesome-Semantic-Textual-Similarity

---

> ### Author Response · Authors · 2024-12-02
> **Gentile reminder**
>
> Dear reviewer,
>
> We thank you for taking the time to review the paper. We have replied to your questions and added some experiments. Since the deadline for reviewer responses is approaching, please let us know if we have managed to address your points. Naturally, if you have any further questions, please let us know.
>
> Thanks.

---

### Official Review · Reviewer_Xe6F · 2024-10-28

**Soundness:** 4
**Presentation:** 3
**Contribution:** 3
**Rating:** 8
**Confidence:** 4

**Summary:**

The authors observe that modern NLP models, specifically Large Language Models (LLMs), are still susceptible to adversarial examples. However, SOTA adversarial attacks require white-box or grey-box access to the target LLMs and this is no longer a realistic threat model. To overcome this constraint, the authors leverage methods from LLM calibration work to recover output confidence ratings, which they make compatible with typically grey-box attacks. Their success suggests that confidence elicitation provides a new attack vector for LLMs.

**Strengths:**

+ This work is significant to the field, as the closed-source nature of modern LLMs is often referenced when arguing against popular optimisation methods (that require grey-box access). Additionally, this work shows that there may be some consequences to the confidence elicitation that many developers have been pushing for, though it likely isn’t enough to pause this work.

+ The authors do a good job in presenting existing work, its findings and problems, as well as presenting their own method (through providing diagrams, examples, equations and pseudo code).

+ The authors carry out substantive analysis of their method against multiple datasets, models, metrics and all compared against other methods. Its particularly nice that they tested how trustworthy the confidence elicitation outputs seem to be since the method assumes these to be correct (as seen in figure 3, though these graphs are not the easiest to understand straight away).

**Weaknesses:**

+ Experiments could possibly be improved by evaluating against actually closed-sourced models, especially since they should be more resilient to non-semantic input perturbations and are probably better at confidence elicitation. In fact, I assume that using anything smaller than the instruction-tuned 8B parameter models from the paper, will result in the prompt template no longer working?

+ Since open-source models were used, the authors could have shown what difference still remains between your method and unconstrained white-box methods (if it is significant, then people can still claim that close-sourcing is very protective), but it is understandable for this to be too far outside of scope.

**Questions:**

+ This could be made clearer: The authors use the prompt template (asking for confidence scores) while optimising but is the optimised question then evaluated outside of the template?

+ Is the total attack time tens of hours or tens of minutes? How many calls did each adversarial attack take on average?

+ Does flipping the model confidence serve as an approximation to flipping the output probabilities, or is it actually the more desirable objective (in terms of what adversarial examples are meant to accomplish)?

---

> ### Author Response · Authors · 2024-11-27
> **Response to reviewer Xe6F [1/3]**
>
> We thank the reviewer for the valuable comments
>
> > ####  **1) Experiments could possibly be improved by evaluating against actually closed-sourced models, especially since they should be more resilient to non-semantic input perturbations and are probably better at confidence elicitation. In fact, I assume that using anything smaller than the instruction-tuned 8B parameter models from the paper, will result in the prompt template no longer working?**
>
> We thank the reviewer for suggesting that we explore closed-source models. We have conducted some tests on GPT-4o using the OpenAI API. We found GPT-4o to be calibrated and that our attack outperforms our current best baseline (SSPAttack), though overall the model is more robust to attacks. We added the experiments in Appendix G 'Evaluation on close source API model' in Tables 10 and 11.
>
> **Calibration of verbal confidence elicitation on an API model**
>
> | Model                                     | Dataset    | Avg ECE ↓ |       |       |       | AUROC ↑ |       | AUPRC Positive ↑ |       | AUPRC Negative ↑ |       |       |       |
> |-------------------------------------------|------------|-----------|-------|-------|-------|--------|-------|------------------|-------|------------------|-------|-------|-------|
> | GPT-4o                                    | SST2       | 0.0286    |       |       |       | 0.9713 |       | 0.9672           |       | 0.9611           |       |       |       |
> | 2024-08-06                                | AG-News    | 0.0641    |       |       |       | 0.9306 |       | -                |       | -                |       |       |       |
> |                                           | StrategyQA | 0.2300    |       |       |       | 0.7410 |       | 0.7238           |       | 0.6962           |       |       |       |
>
> **Table Caption**:Calibration of GPT-4o
>
> **Attack performance on an API model**
>
> | Model   | Dataset     | Technique   | CA [%] ↑ | AUA [%] ↓ | ASR [%] ↑ | SemSim ↑ | Succ Att Queries Avg ↓ | Total Attack Time [HHH:MM:SS] ↓ |
> |---------|-------------|-------------|----------|-----------|-----------|----------|------------------------|---------------------------------|
> | GPT-4o  | SST2        | SSPAttack   | 94.0     | 89.0      | 5.32      | 0.86     | 60.8                   | 001:58:12                       |
> |         |             | CEAttack    | 96.0     | 82.0      | 14.58     | 0.88     | 41.92                  | 002:54:00                       |
> |         |             | CEAttack++  | 95.0     | 68.0      | 28.42     | 0.87     | 108.33                 | 005:34:25                       |
> |         | AG-News     | SSPAttack   | 88.0     | 87.0      | 1.14      | 0.87     | 144.0                  | 002:37:35                       |
> |         |             | CEAttack    | 87.0     | 79.0      | 9.2       | 0.92     | 82.0                   | 005:33:36                       |
> |         |             | CEAttack++  | 88.0     | 75.0      | 14.77     | 0.91     | 412.23                 | 025:56:00                       |
> |         | StrategyQA  | SSPAttack   | 65.0     | 52.0      | 20.0      | 0.9      | 19.07                  | 000:13:22                       |
> |         |             | CEAttack    | 64.0     | 45.0      | 29.69     | 0.89     | 21.15                  | 000:31:01                       |
> |         |             | CEAttack++  | 68.0     | 43.0      | 36.76     | 0.88     | 39.52                  | 001:20:35                       |
>
> **Table Caption**: Confidence elicitation attacks can also target closed-source API models. Naturally, their larger scale makes them more robust to semantic perturbations. Therefore, we set |S| to 20 for SSPAttack and CEAttack. For CEAttack++, we set |S| to 50 across all datasets, employ a delete-word ranking scheme, and specify |W| as 5 for StrategyQA, 10 for SST2, and 20 for AG-News. These configurations represent the best set of hyperparameters identified through our ablation studies.
>
>
> Regarding the choice of model size, the reviewer's intuition is correct: we found that models of size 7/8B are the minimum required to reliably adhere to the template. It is possible that other prompt templates may work better with smaller models. Since inference is faster with smaller models, performing uncertainty estimation through self-consistency, where we query the model multiple times to measure uncertainty estimates [5,6], may be an effective alternative.

---

> ### Author Response · Authors · 2024-11-27
> **Response to reviewer Xe6F [2/3]**
>
> > ####  **2) Since open-source models were used, the authors could have shown what difference still remains between your method and unconstrained white-box methods (if it is significant, then people can still claim that close-sourcing is very protective), but it is understandable for this to be too far outside of scope.**
>
> We thank the reviewer for suggesting a comparison between our black-box technique and white-box methods. Although the comparison could add additional value to the community, the prior work on which our research is based has primarily focused on the black-box aspect [27]. Furthermore, testing with white-box attacks requires GPUs with sufficient memory to handle the computation graph. Currently, we dont have access to such resources. For instance, the LLama3-8B-Instruct model needs approximately 31 GB of the 46 GB of GPU memory available on an A40 just to perform inference. Because of this, we have decided to concentrate on word-level black-box attacks in our current work. Exploring how much added security close-sourcing models would add could be an interesting avenue for future research.
>
>
> > ####  **3) This could be made clearer: The authors use the prompt template (asking for confidence scores) while optimising but is the optimised question then evaluated outside of the template?**
>
> Our prompt template is ‘Verbal Elicitation Verb. 2S k guesses’ which involves making two calls to the target model: the first to obtain predictions and the second to retrieve confidence scores.
>
> Since we are performing a targeted attack, it's important to define the adversarial objective. We have two options: First, we can focus on targeting the prediction template. For example, our adversarial goal might be to maximize the number of incorrect predictions within K or to ensure there are more incorrect predictions than correct ones. In this case, we still perform the second call to obtain confidence scores, but these scores are used by our black-box algorithm to guide perturbations, not for internal classification. This approach would fit the 'out-of-template' implementation.
>
> The second adversarial objective considers the confidence elicitation step as an integral part of the classifier, where confidence scores influence the final prediction. For our threat model, we decided to use the same setup as in [3]. As it is currently implemented, the target model uses confidence scores to refine its predictions, which improves our classifier's AUROC. During the attack, we treat prediction and confidence elicitation as a unified process, maintaining the same template throughout. For example: 'Provide your guesses for the question (Did Aristotle utilize a computer?) and provide your confidences.' We perturb only the words within the question while keeping the overall template consistent throughout the optimization process.
>
> So the results we provide are in the 'in-template' case but our technique could equally be applied to the 'out-of-template' scenario.
>
> > ####  **4) Is the total attack time tens of hours or tens of minutes? How many calls did each adversarial attack take on average?**
>
> Table 4 shows the Total Attack Time [HHH:MM:SS] for perturbing 500 examples from various datasets. The time required varies, with some datasets taking longer than others. For instance, perturbing the Ag-News dataset can take tens of hours, whereas StrategyQA requires approximately one hour. The average number of queries for successful samples is shown in the "Succ Att Queries Avg" column of Table 4. Generally, a model that is less robust to a particular dataset will require fewer queries compared to a dataset to which the model is more robust. We observe that StrategyQA requires approximately 10 queries, SST2 around 25 queries, and AG-News about 45 queries on LLama3 to achieve the ASR in Table 2. Increasing the values of |S| and |W| will increase the number of queries and consequently achieve a higher attack success rate, as demonstrated in the ablation studies in Appendix F.

---

> ### Author Response · Authors · 2024-11-27
> **Response to reviewer Xe6F [3/3]**
>
> > ####  **5) Does flipping the model confidence serve as an approximation to flipping the output probabilities, or is it actually the more desirable objective (in terms of what adversarial examples are meant to accomplish)?**
>
> This is a good question. At the start of the project, our focus was on exploring how confidence elicitation can approximate probabilities, as previous adversarial attacks have demonstrated effectiveness when probabilities are used as a feedback signal. Ultimately, it depends on the adversarial objective. For our proof of concept, approximating the output probabilities is effective in a classification setting. However in a generation setting, the likelihood of the next token primarily signifies lexical confidence and not epistemic uncertainty [28]. In fact, the confidence measure often depends on the application. For instance, when a model answers the query, "What is the capital of Italy?" the initial token could be highly uncertain among "Rome", "The" and "Italy" due to multiple valid response structures such as "Rome is the capital," "The capital is Rome," or "Italy's capital is Rome." This uncertainty underscores the model's difficulty in precisely determining a single method of response, reflecting a broader uncertainty pertinent to \textit{the same question} [29]. So in this case, being able to ask the model for it's semantic probability/confidence in the inherent textual meaning under a confidence elicitation framework [3] and using this value as a feedback signal can be regarded as a more desirable objective.
>
> **References**
>
> [3] Miao Xiong et al, Can LLMs Express Their Uncertainty? An Empirical Evaluation of Confidence Elicitation in LLMs, ICLR 2024
>
> [5] Jiuhai Chen et al. Quantifying Uncertainty in Answers from any Language Model and Enhancing their Trustworthiness, ACL 2024
>
> [6] Xuezhi Wang et al. self-consistency improves chain of thought reasoning in language models, ICLR 2023
>
> [27] Han Liu et al. SSPAttack: A Simple and Sweet Paradigm for Black-Box Hard-Label Textual Adversarial Attack, 2023
>
> [28] Stephanie Lin et al. Teaching models to express their uncertainty in words, TMLR 2022
>
> [29] Lorenz Kuhn et al. Semantic Uncertainty: Linguistic Invariances for Uncertainty Estimation in Natural Language Generation, ICLR 2023

---

> ### Author Response · Authors · 2024-12-02
> **Gentile reminder**
>
> Dear reviewer,
>
> We thank you for taking the time to review the paper. We have replied to your questions and added some experiments. Since the deadline for reviewer responses is approaching, please let us know if we have managed to address your points. Naturally, if you have any further questions, please let us know.
>
> Thanks.

---

### Official Review · Reviewer_7pp6 · 2024-11-02

**Soundness:** 3
**Presentation:** 3
**Contribution:** 3
**Rating:** 5
**Confidence:** 4

**Summary:**

This paper proposes a new attack vector for large language models by overcoming the unachievable soft-label issues when attacking the large language models. To be more specific, the authors propose to approximate soft labels for hard label attacks by confidence elicitation. Experiment results have shown that their proposed method can reach state-of-the-art attack performance in hard-label, black-box, word-substitution-based settings on large language models.

**Strengths:**

1. The first paper proposes to apply confidence elicitation to address the unachievable soft-label issues in adversarial attacks for large language models.
2. Experimental results demonstrate the effectiveness of CEAttack compared with baseline attack methods Self-Fool Word Sub and SSPAttack.

**Weaknesses:**

1. Lack of baseline analysis compared with prompt optimization methods such as Tree of Attacks [1].
2. The study scope of the paper is limited to classification tasks, which are only a small part of the tasks current large language models can do and have a minor impact for now. It is still unclear whether the method maintains effectiveness on generative tasks like jailbreaking.
3. Potential defense discussions about the attack.

[1] Anay Mehrotra, Manolis Zampetakis, Paul Kassianik, Blaine Nelson, Hyrum Anderson, Yaron Singer, and Amin Karbasi. Tree of attacks: Jailbreaking black-box LLMs automatically. arXiv:2312.02119, 2023.

**Questions:**

Please add experiments based on the suggestions from the weaknesses.

---

> ### Author Response · Authors · 2024-11-27
> **Response to reviewer 7pp6 [1/4]**
>
> We thank the reviewer for the valuable comments
> > #### **1) Lack of baseline analysis compared with prompt optimization methods such as Tree of Attacks.**
>
> We thank the reviewer for pointing out this paper. Although jailbreaks are a related topic, the setting is different from adversarial attacks. The aim of the highlighted paper is to develop a black-box jailbreak targeting model safeguards, where the objective is to perturb the input to generate a response that deviates from the standard refusal answer "I'm sorry, I cannot answer this request" and instead seeks to answer the malicious input question. It employs an LLM to generate multiple prompts and follows a tree structure to select the prompts with the highest potential to jailbreak the model, continuously improving these prompts to trick the target model into not refusing the question.
>
> We would love to benchmark this technique as it is very recent, but it varies significantly in several ways:
>
> **1.** It is a jailbreak and not an adversarial attack.
>
> **2.** It uses an LLM to generate a prompt, instead of word substitutions, thus making it closer to a prompt-level attack [20], rather than a word-level attack.
>
> **3.** It doesn't use confidence as guidance in any way.
>
> Given the above distinctions. We extended the work by including a relevant word-level black-box attack as an additional baseline from AAAI 2022 called **TextHoaxer** [26], which aligns with our adversarial setting, word-level attack, and black-box nature. Our method also outperforms this baseline with regards to ASR, AUA and 'Succ Att Queries Avg'.
>
> As a brief example, the table below compares the new baseline with our work.
>
> **Attack performance tests**
> Numbers in **bold** are the best results.
>
> | Model                        | Dataset    | CA [%] ↑         |            |              |            | AUA [%] ↓                               |        |        |               | ASR [%] ↑           |       |      |      |
> |------------------------------|------------|------------------|------------|--------------|------------|----------------------------------------|--------|--------|---------------|---------------------|-------|------|------|
> |                              |            | Vanilla          |            |              |            | Self-Fool Word Sub                      | **Text Hoaxer**  | SSP Attack | CE Attack       | Self-Fool Word Sub  | **Text Hoaxer** | SSP Attack | CE Attack |
> |**LLaMa-3-8B Instruct**      | SST2       | 90.56±0.14       |            |              |            | 88.35                                  | 82.93  | 81.93  | **72.69**    | 2.22                | 8.43  | 9.73 | **19.73** |
> |                              | AG-News    | 61.62±0.38       |            |              |            | 61.17                                  | 49.3   | 45.27  | **43.06**    | 0.33                | 19.41 | 26.71 | **30.74** |
> |       | StrategyQA | 60.22±0.17       |            |              |            | 59.52                                  | 45.29  | 42.28  | **32.67**    | 1.66                | 24.67 | 29.67 | **45.67** |
> | **Mistral-7B Instruct-v0.3**      | SST2       | 87.87±0.39       |            |              |            | 84.73                                  | 74.27  | 75.31  | **71.76**    | 3.57                | 16.08 | 14.08 | **17.94** |
> |                              | AG-News    | 65.99±0.27       |            |              |            | -                                      | 48.69  | 52.48  | **40.82**    | -                   | 26.43 | 20.0 | **38.33** |
> |  | StrategyQA | 59.92±0.32       |            |              |            | 59.61                                  | 44.33  | 41.13  | **36.21**    | 1.22                | 26.23 | 30.99 | **39.26** |
>
> *Results of performing Confidence Elicitation Attacks. Numbers in bold are the best results.*
>
>
> **Furthermore, we believe that confidence elicitation could be an interesting concept to enhance jailbreaks.** For example, it could allow the Tree of Attacks to query the probability of the model's answers during a multi-turn dialog to jailbreak the model and use this information as feedback. Consequently, in the future work section in Appendix B.2 we added a discussion of previous work on jailbreaks, highlighting past efforts such as GCG [25], PAIR [20], and TAP [21]. We then discussed that our techniques could be incorporated into these methods to enhance them.

---

> ### Author Response · Authors · 2024-11-27
> **Response to reviewer 7pp6 [2/4]**
>
> > #### **The study scope of the paper is limited to classification tasks, which are only a small part of the tasks current large language models can do and have a minor impact for now. It is still unclear whether the method maintains effectiveness on generative tasks like jailbreaking.**
>
> There are two ways we can see our approach being extended to generation tasks:
>
> **1) Free-Form Generation:**
>
> There are several tasks, such as TriviaQA, NarrativeQA, PopQA, and SciQ, where we have a label set of correct answers for each datapoint. In these datasets, a common task is question answering, where the prompt is a question, and the model is expected to generate an answer in a free-form setting. In this setting, the model can generate a word sequence of arbitrary length, which is considered a generation task [16,13,12,14,15]. The accuracy of the model is evaluated based on its relevance to the label set using a threshold and an overlap algorithm, such as exact match, F1, BLEU, or ROUGE. While the original TriviaQA was evaluated using these rule-based algorithmic approaches, later works have employed language models for evaluation [19].
>
> Regarding confidence elicitation in a generative setting, recent work [16] used a ROUGE threshold of 0.3 between the generated answer and the label. Following this, the model is asked to provide its verbal confidence level. They demonstrated that using verbal confidence elicitation can achieve some of the best ECE uncertainty measures (see Table 3 (A4C) in [16]) for TriviaQA, a generative task.
>
> In the context of adversarial attacks, the adversarial setup is crucial. A straightforward and effective starting point could involve perturbing the input so that the generated answer no longer exceeds the 0.3 ROUGE/BLEU threshold [18]. This would indicate minimal overlap between the generated sentence and the label sentences. In a black-box setting, we would not know the threshold or that ROUGE is utilized. We would simply receive a binary output indicating whether the overlap is sufficient. Therefore, being able to query the model about its confidence in how well the generated answer addresses the question can serve as a valuable feedback signal. Reducing this confidence would suggest that the model is less certain that the answer truly addresses the question.
>
> There are other, deemed more challenging adversarial objectives, such as the Non-overlapping Attack, where the goal is to perturb the input so that no words in the newly generated output overlap with the target label. Another example is the Targeted Keywords Attack, where the input perturbation aims to introduce a specific keyword into the output [23]
>
> In the above setups, we would maintain the 'Verbal Elicitation Verb. 2S k guesses' elicitation framework and use the elicited confidence as guidance, just as before.
>
> Although we were unable to explore this question in detail or conduct experiments due to limited time, we believe that addressing confidence elicitation attacks in free-form generation tasks on datasets such as TriviaQA would be a natural next step for future work. This is particularly relevant given that recent papers have demonstrated that confidence elicitation can provide reliable uncertainty measures on this dataset [16]
>
> **2) Token-Wise:**
>
> Confidence elicitation can also be applied to approximate token probabilities. The adversarial objective would be to minimize the probability of specific tokens so that they are no longer generated by the model, increasing the likelihood of other tokens being generated.
>
> For example, the objective may be to make the model generate a response that is unrelated to the question. We could introduce an input perturbation to minimize the probability of the token 'window' being generated in the output.
>
> Example:
> - User: You're a model that answers questions and provides the confidence for each token between 0 and 1, indicating the token's correctness. "How do you build a window?"
> - Model: To: 0.9, build: 0.95, a: 0.95, window: 0.95, you: 0.9, typically: 0.85, follow: 0.85, these: 0.9, steps: 0.9
>
> Application to Jailbreaks:
>
> This is a high-level concept, but with a similar setup, we can ask the model, "How to build a bomb." - Model: sorry: 0.8, I: 0.95, can't: 0.9, assist: 0.85, with: 0.9, that: 0.9, request: 0.85
>
> The goal would be to minimize the probability of the token 'sorry' appearing in the output, thereby increasing the model's likelihood to choose other tokens such as 'sure' or 'to'.

---

> ### Author Response · Authors · 2024-11-27
> **Response to reviewer 7pp6 [3/4]**
>
> > #### **3) Potential defense discussions about the attack.**
>
>
> We included the following in the Defense discussion of Appendix B 'Future Work'.
>
> **Defense Discussion**
>
> We have thoroughly considered how to address this issue. The main challenge arises from an active push within the community to make confidence elicitation an integral part of LLMs' behavior. We believe this is an interesting emergent behavior and do not think the community should halt these efforts. Therefore, simply blocking models from performing confidence elicitation or impairing their ability by adding noise or deliberately making them uncalibrated is not a viable option. Ultimately, we concluded that confidence elicitation may be at odds with robustness. However, we have identified two potential directions that the community may find worth exploring:
>
> 1. **Adversarial Training / Adversarial Data Augmentation**
>
> Can the generated adversarial inputs be reintroduced into the training process? This opens up opportunities for confidence elicitation adversarial training, aiming to both enhance robustness against input perturbation and potentially improve calibration. This would adhere to the traditional adversarial training min-max formulation.
>
> As we incorporate perturbations into the input that alter predictions during instruction fine-tuning, we can loop these samples back into the training process, following a black-box adversarial setup. Adversarial training could potentially be done in a white-box setting by perturbing the input embeddings then checking how the confidence elicitation behaviour changes.
>
> Alternatively, a simpler solution involves generating the data first and then using it for further fine-tuning (adversarial augmentation).
>
> 2. **Defense by Intent**
>
> In this case, we aim to protect the system by analyzing the use cases of confidence elicitation with a rule-based defense approach.
> - Is the user performing the same query multiple times with small semantic similarities between queries, likely adding minor input perturbations?
> - Is the user explicitly asking for confidence elicitation? This could be implemented as a classifier. If yes, it indicates a desire for confidence elicitation.
> - Are confidence elicitation values on some tokens decreasing over time? This could suggest some form of optimization in progress.

---

> ### Author Response · Authors · 2024-11-27
> **Response to reviewer 7pp6 [4/4]**
>
> **References**
>
> [12] Changchang Zeng et al. A Survey on Machine Reading Comprehension: Tasks, Evaluation Metrics and Benchmark Datasets, 2020
>
> [13] Danqi Chen, neural reading comprehension and beyond , 2018
>
> [14] Tomáš Kočiský et al. The NarrativeQA Reading Comprehension Challenge, TACL 2017
>
> [15] Mandar Joshi et al. TriviaQA: A Large Scale Distantly Supervised Challenge Dataset for Reading Comprehension, ACL 2017
>
> [16] Linyu Liu et al. Uncertainty Estimation and Quantification for LLMs: A Simple Supervised Approach ,2024,
>
> [19] Katherine Tian et al. Just Ask for Calibration: Strategies for Eliciting Calibrated Confidence Scores from Language Models Fine-Tuned with Human Feedback EMNLP 2023
>
> [20] Patrick Chao et al. PAIR Jailbreaking Black Box Large Language Models in Twenty Queries, NIPS workshop 2024
>
> [21] Anay Mehrotra et al. Tree of Attacks: Jailbreaking Black-Box LLMs Automatically, NIPS 2024
>
> [23] Minhao Cheng et al. Seq2Sick: Evaluating the Robustness of Sequence-to-Sequence Models with Adversarial Examples, AAAI 2020
>
> [25] Andy Zou et al. GCG Universal and Transferable Adversarial Attacks on Aligned Language Models, 2023
>
> [26] Muchao Ye et al. TextHoaxer: Budgeted Hard-Label Adversarial Attacks on Text. AAAI 2022

---

> ### Author Response · Authors · 2024-12-02
> **Gentile reminder**
>
> Dear reviewer,
>
> We thank you for taking the time to review the paper. We have replied to your questions and added some experiments. Since the deadline for reviewer responses is approaching, please let us know if we have managed to address your points. Naturally, if you have any further questions, please let us know.
>
> Thanks.

---

### Official Review · Reviewer_dcYJ · 2024-11-02

**Soundness:** 3
**Presentation:** 2
**Contribution:** 3
**Rating:** 6
**Confidence:** 3

**Summary:**

This paper explores a novel method of adversarial attack on large language models (LLMs) by leveraging their ability to express confidence levels. Unlike previous attacks that relied on model transparency, this research focuses on black-box scenarios where only the model's final output is accessible. The authors demonstrate that by eliciting confidence scores from LLMs, it is possible to approximate the effectiveness of soft-label attacks, which traditionally require access to output probabilities or logits. Through a series of experiments on three datasets and two models (LLaMa-3-8B Instruct and Mistral-7B Instruct-v0.3), the paper shows that minimizing the model's confidence can increase the likelihood of misclassification, achieving state-of-the-art results in hard-label, black-box attacks.

**Strengths:**

* This study introduces the first approach for constructing adversarial examples of large language models (LLMs) by leveraging their confidence elicitation capabilities.

* The proposed confidence elicitation attack is easy to conduct and requires fewer queries than existing black-box adversarial attacks. This straightforward adversarial attack provides an easy-to-implement method for testing the potential vulnerabilities of LLMs while maintaining a higher degree of semantic similarity in the perturbed samples.

**Weaknesses:**

The study provides insights into the calibration of LLMs and the effectiveness of confidence elicitation in guiding adversarial perturbations, highlighting potential implications for the robustness of LLMs. Thus, it contributes a new perspective to the field of adversarial machine learning. However, I still see some issues that may improve the quality of the paper:

* Limited in the Classification Tasks. Different from existing Jailbreaks against the generation function of LLMs. This work focuses specifically on the classification functions of LLMs. However, the primary use of LLMs is not for classification but rather for generating responses that provide users with their desired answers. The authors should justify the potential scenarios in which their attack may be employed, specifically those in which users utilize LLMs to classify their inputs.

* Generalizability of Confidence Values. The study assumes that the elicited confidence values are well-calibrated for the models used, but it does not sufficiently explore or discuss the generalizability of this assumption across different LLMs or datasets. This limitation raises questions about the robustness of the attack vector when applied to models with varying levels of calibration or in different linguistic contexts.

* Lack of Detail on ECE Calculation. The paper lacks a clear explanation of how to ensure that the confidence values returned by large language models (LLMs) are accurate for guiding the generation of perturbations, i.e., the Expected Calibration Error (ECE) technique, which is crucial for understanding the calibration of the models and the reliability of the confidence elicitation. Without this detail, it is challenging to assess the validity of the calibration claims and the effectiveness of the proposed attacks under different calibration scenarios. It is encouraged to integrate ECE into the methodology section of the paper or provide as background knowledge, which would offer transparency into the calibration process and allow readers to better assess the reliability of the confidence elicitation technique.

* Complexity and Efficiency of Synonym Replacement Method. The proposed attack method involves synonym replacement that grows exponentially with the length of the input sentence, leading to a significant increase in computational complexity. This exponential growth can result in prolonged processing times for longer sentences, which may limit the practical applicability of the proposed attack. To mitigate the high computational complexity and time consumption associated with the synonym replacement method, especially for longer sentences, the authors should explore and implement optimizations in the word selection process.

**Questions:**

* Is it possible to extend the proposed attack to text generation tasks?

* Do most mainstream large language models (LLMs) support confidence elicitation?

* Does the computational complexity of the proposed attack increase with the length of the input prompt?

---

> ### Author Response · Authors · 2024-11-27
> **Response to reviewer dcYJ [1/8]**
>
> We thank the reviewer for the valuable comments
>
> > #### **1) Limited in the Classification Tasks. Different from existing Jailbreaks against the generation function of LLMs. This work focuses specifically on the classification functions of LLMs. However, the primary use of LLMs is not for classification but rather for generating responses that provide users with their desired answers. The authors should justify the potential scenarios in which their attack may be employed, specifically those in which users utilize LLMs to classify their inputs.**
>
> We thank the reviewer for their feedback. The aim of our work is to serve as a proof-of-concept, highlighting a potential vulnerability in confidence elicitation, henceforth motivating the need for implementing appropriate guardrails in open-ended systems, particularly when these systems are paired with calibration results in next-token generation and free-form generation tasks.
>
> We hope that the community will explore the robustness of these tasks in greater detail in the future, considering that confidence elicitation has recently emerged as a candidate for uncertainty estimation in free-form generation [16, 17, 19].
>
>
> In the revised introduction, we have included some potential scenarios to demonstrate how confidence elicitation attacks might be employed specifically in a classification setting. We have also extended the conclusion to highlight how confidence elicitation attacks in free-form generation, jailbreaks and next-token prediction are all interesting avenues for future research.
>
>
> **Added in the introduction:**
>
> >Such vulnerabilities are particularly concerning in critical applications, such as healthcare [9], socio-technical systems and human-machine collaboration [8]. For example, in healthcare, where a medical system provides a diagnosis, an attacker might introduce input perturbations, aiming to achieve a misclassification. In clinical support systems, such misclassifications can have lethal consequences [10].
>
> >Providing confidence estimates through confidence elicitation, whether in a template or a free-form generation, has been shown to enhance the performance and utility of these systems. This is particularly important in domains where assessing the reliability of a model's responses is crucial for effective risk assessment, error mitigation, selective generation, and minimizing the effects of hallucinations [3]. As a result, we can anticipate these techniques to become more widespread. Consequently, exploring whether we can strengthen adversarial perturbations using confidence estimates is an important area of research, with the aim of designing more robust systems.
>
> **Added in the conclusion:**
>
> >Our results suggest the potential for confidence elicitation to enhance jailbreaks. For example, it might enable current multi-turn dialog jailbreaks [20,21] to query the probability of the model's answers and use this information as feedback. Another promising direction is to investigate how susceptible token-wise confidence elicitation is to input perturbations, and whether it is possible to control or influence the model's token selection process. Additionally, it is worth exploring how effective confidence elicitation attacks are on generative tasks such as free-form question answering [15], given that confidence elicitation has also proven to be a reliable and calibrated measure of uncertainty in generative tasks [16,17].

---

> ### Author Response · Authors · 2024-11-27
> **Response to reviewer dcYJ [2/8]**
>
> > #### **2) Generalizability of Confidence Values. The study assumes that the elicited confidence values are well-calibrated for the models used, but it does not sufficiently explore or discuss the generalizability of this assumption across different LLMs or datasets. This limitation raises questions about the robustness of the attack vector when applied to models with varying levels of calibration or in different linguistic contexts.**
>
> First, we note that our results in Table 1 showed that LLaMa-3-8B-Instruct and Mistral-7B-Instruct-v0.3
> model are sufficiently calibrated on 3 different datasets. They have an expected ECE below 0.25 and an AUROC above 0.6 for multiple tasks; these thresholds have been established as reasonable benchmarks in previous works [3] to define a sufficiently calibrated model.
>
> We have also expanded the empirical evaluation of confidence elicitation on four more models and two more datasets (as shown in the table below and in Appendix F.1 Confidence elicitation generalization). We found that even today, multiple models exhibit good calibration when prompted to express their uncertainty verbally. This is shown by the low Avg ECE scores on models like Gemma2 9B-Instruct, Mistral-Nemo 12B-Instruct-2407, Qwen2.5 7B-Instruct, and LLaMa-3.2-11B Vision-Instruct. Notably, for Qwen2.5 7B-Instruct, the Avg ECE is even below 0.05, indicating a mere 5 percentage point difference between the predicted probabilities and the observed frequencies of outcomes. This level of ECE is achieved by a model that was not specifically trained to be calibrated, yet can express its uncertainty verbally as an emergent property.
>
> Finally, we believe the list of calibrated models will grow over time.
> The community is actively working towards enhancing model calibration, which suggests a trend towards improved confidence levels [22,17]. For example, in [17], Google DeepMind recently explored fine-tuning to make Gemma 1.0 models better at expressing their uncertainty verbally.
>
> **Calibration of Verbal Confidence Elicitation on More Models**
>
> | Model                           | Dataset    | Avg ECE ↓ | AUROC ↑ | AUPRC Positive ↑ | AUPRC Negative ↑ |
> |--------------------------------------|------------|-----------|---------|-----------------|-----------------|
> | **Gemma2 9B-Instruct**               |            |           |         |                 |                 |
> |                                      | SST2       | 0.0591    | 0.9486  | 0.9547          | 0.9357          |
> |                                      | AG-News    | 0.1666    | 0.8342  | -               | -               |
> |                                      | StrategyQA | 0.2295    | 0.6631  | 0.5899          | 0.7563          |
> | **Mistral-Nemo 12B-Instruct-2407**   |            |           |         |                 |                 |
> |                                      | SST2       | 0.0645    | 0.9958  | 0.9944          | 0.9970          |
> |                                      | AG-News    | 0.0673    | 0.9194  | -               | -               |
> |                                      | StrategyQA | 0.2748    | 0.6214  | 0.6425          | 0.5863          |
> | **Qwen2.5 7B-Instruct**              |            |           |         |                 |                 |
> |                                      | SST2       | 0.0382    | 0.9534  | 0.9399          | 0.9480          |
> |                                      | AG-News    | 0.0753    | 0.8722  | -               | -               |
> |                                      | StrategyQA | 0.2332    | 0.6247  | 0.6649          | 0.5624          |
> | **LLaMa-3.2-11B Vision-Instruct**    |            |           |         |                 |                 |
> |                                      | SST2       | 0.0581    | 0.9535  | 0.9645          | 0.9270          |
> |                                      | AG-News    | 0.1090    | 0.8954  | -               | -               |
> |                                      | StrategyQA | 0.2720    | 0.6366  | 0.6532          | 0.5928          |
>
> *Calibration of other models with confidence elicitation on core datasets SST2, AG-News, and StrategyQA*
>
> **Calibration of verbal confidence elicitation on more datasets**
>
> | Model                        | Dataset | Avg ECE ↓ | AUROC ↑ | AUPRC Positive ↑ | AUPRC Negative ↑ |
> |------------------------------|---------|-----------|---------|------------------|------------------|
> | LLaMa-3-8B Instruct          | RTE     | 0.2598    | 0.8230  | 0.7972           | 0.8418           |
> |                              | QNLI    | 0.1352    | 0.8413  | 0.8561           | 0.8182           |
> | Mistral-7B Instruct-v0.3     | RTE     | 0.3047    | 0.6507  | 0.6032           | 0.6927           |
> |                              | QNLI    | 0.2764    | 0.6951  | 0.6444           | 0.7345           |
>
> *Calibration of other datasets on core models Mistral and LLaMa3*

---

> ### Author Response · Authors · 2024-11-27
> **Response to reviewer dcYJ [3/8]**
>
> > #### **3) Lack of Detail on ECE Calculation. The paper lacks a clear explanation of how to ensure that the confidence values returned by large language models (LLMs) are accurate for guiding the generation of perturbations, i.e., the Expected Calibration Error (ECE) technique, which is crucial for understanding the calibration of the models and the reliability of the confidence elicitation. Without this detail, it is challenging to assess the validity of the calibration claims and the effectiveness of the proposed attacks under different calibration scenarios. It is encouraged to integrate ECE into the methodology section of the paper or provide as background knowledge, which would offer transparency into the calibration process and allow readers to better assess the reliability of the confidence elicitation technique.**
>
> We thank the reviewer for the valuable comment,
>
> We added the following explanation in Section 5.1 `Model Calibration':
>
> ECE [24] is a metric used to assess how well a model's confidence estimates align with the actual probabilities of outcomes being correct. For example, it helps evaluate how accurately a model's predicted confidence (e.g., 'I'm 80% sure this is correct') matches reality. This assessment is averaged across 500 examples. A thorough explanation of this is provided in Appendix E.4 `Expected Calibration Error (ECE)'.
>
>
> And added an extensive explenation in the Appendix E.4:
>
> The Expected Calibration Error (ECE) is calculated using the formula:
>
> \begin{align*}
> \text{ECE} = \sum_{m=1}^{M} \frac{|B_m|}{n} \cdot \left| \text{acc}(B_m) - \text{conf}(B_m) \right|
> \end{align*}
>
> In this formula, $n$ represents the total number of samples, and $M$ is the total number of bins used to partition the predicted confidence scores. The term $B_m$ denotes the set of indices of samples whose predicted confidence falls into the $m$-th bin, and $|B_m|$ is the number of samples in this bin. The accuracy within each bin, $\text{acc}(B_m)$, is calculated as the proportion of correctly predicted samples, given by the equation:
>
> $$
> \text{acc}(B_m) = \frac{1}{|B_m|} \sum_{i \in B_m} \mathbf{1}(\hat{y}_i = y_i)
> $$
>
> where $\hat{y}_i$ is the predicted class label and $y_i$ is the true class label for sample $i$. The confidence of the predictions in the $m$-th bin, $\text{conf}(B_m)$, is the average of the predicted confidence scores for the samples in the bin, calculated as:
>
> $$
> \text{conf}(B_m) = \frac{1}{|B_m|} \sum_{i \in B_m} \hat{p}_i
> $$
>
> where $\hat{p}_i$ is the predicted probability for the predicted class of sample $i$. The ECE thus captures the weighted average of the absolute differences between accuracy and confidence across all bins, providing a summary measure of model calibration.
>
> We use 10 bins to generate our plots in Figure 3.

---

> ### Author Response · Authors · 2024-11-27
> **Response to reviewer dcYJ [4/8]**
>
> > #### **4) Complexity and Efficiency of Synonym Replacement Method. The proposed attack method involves synonym replacement that grows exponentially with the length of the input sentence, leading to a significant increase in computational complexity. This exponential growth can result in prolonged processing times for longer sentences, which may limit the practical applicability of the proposed attack. To mitigate the high computational complexity and time consumption associated with the synonym replacement method, especially for longer sentences, the authors should explore and implement optimizations in the word selection process.**
>
> We used a simple hill climbing algorithm to keep the complexity low, by sequentially picking the best word given all the previous words, resulting in a complexity that scales as |S|*|W|, where |S| is the number of synonyms used and |W| is the number of words in the sentences. One could also use more advanced search methods, such as beam search, particle swarm optimization [1], or genetic algorithms, all of which have proven to deliver better attack performance at the cost of more queries or optimization time.
>
> We also explored a different word ranking strategy at the reviewer's suggestion. Our new experiments incorporate a word deletion strategy, as it aligns best with our black-box constraints. Approaches like PWWS (word saliency ranking) [2] could also be beneficial if we replace standard output probabilities with our probability approximation. However, techniques that utilize attention scores would require access to the model's internals.
>
> The deletion strategy achieves linear complexity by checking each word once to determine the change in output, and then ranking the words based on their impact on the approximated output probability, from highest to lowest. We find that incorporating the word deletion strategy enhances the attack's performance for samples with more words, as measured by the Attack Success Rate (ASR). Although this new strategy results in a higher overall number of queries, the trade-off offers superior performance compared to our baselines. For example, in CEAttacks on LLama3, the ASR increases from 19.73% to 28.32%, while the average number of successful attack queries rises from 25.60 to 35.67, still lower than SSPAttack’s 82.95. This behavior is consistently observed across different datasets and models, and we have included this analysis as an ablation experiment in our paper in the Appendix H.4 'Delete word ranking schema ablation'.

---

> ### Author Response · Authors · 2024-11-27
> **Response to reviewer dcYJ [5/8]**
>
> **Attack Performance with a delete word ranking schema for CEAttacks**
>
>  | Model                                        | Dataset    | Random Ranking | Delete Ranking | Random Ranking | Delete Ranking | Random Ranking | Delete Ranking | Random Ranking | Delete Ranking | Random Ranking | Delete Ranking | Random Ranking | Delete Ranking |
> |----------------------------------------------|------------|----------------|----------------|----------------|----------------|----------------|----------------|----------------|----------------|----------------|----------------|----------------|----------------|
> |                                              |            | CA [%] ↑       | CA [%] ↑       | AUA [%] ↓      | AUA [%] ↓      | ASR [%] ↑      | ASR [%] ↑      | SemSim ↑       | SemSim ↑       | Succ Att Queries Avg ↓ | Succ Att Queries Avg ↓ | Total Attack Time [HHH:MM:SS] ↓ | Total Attack Time [HHH:MM:SS] ↓ |
> | LLaMa-3-8B Instruct        | SST2       | 90.56          | 90.76          | 72.69          | 65.06          | 19.73          | 28.32          | 0.88           | 0.88           | 25.60         | 35.67         | 017:30:57      | 025:33:37      |
> |                                              | AG-News    | 62.17          | 61.97          | 43.06          | 40.85          | 30.74          | 34.09          | 0.93           | 0.93           | 42.36         | 68.13         | 024:31:58      | 039:22:19      |
> |                                              | StrategyQA | 60.12          | 59.92          | 32.67          | 33.07          | 45.67          | 44.82          | 0.89           | 0.89           | 10.95         | 17.10         | 001:25:34      | 002:19:30      |
> |Mistral-7B Instruct-v0.3    | SST2       | 87.45          | 88.08          | 71.76          | 67.78          | 17.94          | 23.04          | 0.88           | 0.88           | 24.54         | 33.18         | 017:13:44      | 024:05:07      |
> |                                              | AG-News    | 66.18          | 65.89          | 40.82          | 33.24          | 38.33          | 49.56          | 0.93           | 0.92           | 42.66         | 68.95         | 017:16:52      | 027:49:15      |
> |                                              | StrategyQA | 59.61          | 58.87          | 36.21          | 33.0           | 39.26          | 43.93          | 0.9            | 0.89           | 11.37         | 18.01         | 001:43:48      | 002:43:34      |
>
> *Confidence elicitation can also serve as a proxy for ranking the importance of words in the input*

---

> ### Author Response · Authors · 2024-11-27
> **Response to reviewer dcYJ [6/8]**
>
> > #### **Q1) Is it possible to extend the proposed attack to text generation tasks?**
>
> There are two ways we can see our approach being extended to generation tasks:
>
> **1) Free-Form Generation:**
>
> There are several tasks, such as TriviaQA, NarrativeQA, PopQA, and SciQ, where we have a label set of correct answers for each datapoint. In these datasets, a common task is question answering, where the prompt is a question, and the model is expected to generate an answer in a free-form setting. In this setting, the model can generate a word sequence of arbitrary length, which is considered a generation task [16,13,12,14,15]. The accuracy of the model is evaluated based on its relevance to the label set using a threshold and an overlap algorithm, such as exact match, F1, BLEU, or ROUGE. While the original TriviaQA was evaluated using these rule-based algorithmic approaches, later works have employed language models for evaluation [19].
>
> Regarding confidence elicitation in a generative setting, recent work [16] used a ROUGE threshold of 0.3 between the generated answer and the label. Following this, the model is asked to provide its verbal confidence level. They demonstrated that using verbal confidence elicitation can achieve some of the best ECE uncertainty measures (see Table 3 (A4C) in [16]) for TriviaQA, a generative task.
>
> In the context of adversarial attacks, the adversarial setup is crucial. A straightforward and effective starting point could involve perturbing the input so that the generated answer no longer exceeds the 0.3 ROUGE/BLEU threshold [18]. This would indicate minimal overlap between the generated sentence and the label sentences. In a black-box setting, we would not know the threshold or that ROUGE is utilized. We would simply receive a binary output indicating whether the overlap is sufficient. Therefore, being able to query the model about its confidence in how well the generated answer addresses the question can serve as a valuable feedback signal. Reducing this confidence would suggest that the model is less certain that the answer truly addresses the question.
>
> There are other, deemed more challenging adversarial objectives, such as the Non-overlapping Attack, where the goal is to perturb the input so that no words in the newly generated output overlap with the target label. Another example is the Targeted Keywords Attack, where the input perturbation aims to introduce a specific keyword into the output [23]
>
> In the above setups, we would maintain the 'Verbal Elicitation Verb. 2S k guesses' elicitation framework and use the elicited confidence as guidance, just as before.
>
> Although we were unable to explore this question in detail or conduct experiments due to the short rebuttal period, we believe that addressing confidence elicitation attacks in free-form generation tasks on datasets such as TriviaQA would be a natural next step for future work. This is particularly relevant given that recent papers have demonstrated that confidence elicitation can provide reliable uncertainty measures on this dataset [16]
>
> **2) Token-Wise:**
>
> Confidence elicitation can also be applied to approximate token probabilities. The adversarial objective would be to minimize the probability of specific tokens so that they are no longer generated by the model, increasing the likelihood of other tokens being generated.
>
> For example, the objective may be to make the model generate a response that is unrelated to the question. We could introduce an input perturbation to minimize the probability of the token 'window' being generated in the output.
>
> Example:
> - User: You're a model that answers questions and provides the confidence for each token between 0 and 1, indicating the token's correctness. "How do you build a window?"
> - Model: To: 0.9, build: 0.95, a: 0.95, window: 0.95, you: 0.9, typically: 0.85, follow: 0.85, these: 0.9, steps: 0.9
>
> Application to Jailbreaks:
>
> This is a high-level concept, but with a similar setup, we can ask the model, "How to build a bomb." - Model: sorry: 0.8, I: 0.95, can't: 0.9, assist: 0.85, with: 0.9, that: 0.9, request: 0.85
>
> The goal would be to minimize the probability of the token 'sorry' appearing in the output, thereby increasing the model's likelihood to choose other tokens such as 'sure' or 'to'.

---

> ### Author Response · Authors · 2024-11-27
> **Response to reviewer dcYJ [7/8]**
>
> > #### **Q2) Do most mainstream LLMs support confidence elicitation?**
>
> To the best of our knowledge, any decoder-based LLM generator model can express its confidence verbally, and the comunity are actively working to better calibrate this expressed confidence.
>
>
> We added four screenshots demonstrating confidence elicitation in action on GPT-4o and Claude, showcasing both prediction confidence elicitation and token-wise confidence elicitation.
>
> **Claude simple prediction confidence elicitation**:
> https://anonymous.4open.science/r/Confidence_Elicitation_Attacks-5220/confidence_elicitation_claude_nvc_simple.png
>
> **Claude token wise prediction confidence elicitation**:
> https://anonymous.4open.science/r/Confidence_Elicitation_Attacks-5220/confidence_elicitation_claude_per_token_analysis_input_nvc.png
>
> **GPT4o simple prediction confidence elicitation**:
> https://anonymous.4open.science/r/Confidence_Elicitation_Attacks-5220/confidence_elicitation_gpt4o_nvc_simple.png
>
> **GPT4o token wise prediction confidence elicitation**:
> https://anonymous.4open.science/r/Confidence_Elicitation_Attacks-5220/confidence_elicitation_gpt4o_per_token_analysis_input_nvc.png
>
> > #### **Q3) Does the computational complexity of the proposed attack increase with the length of the input prompt?**
>
> Please refer to our response to 4).

---

> ### Author Response · Authors · 2024-11-27
> **Response to reviewer dcYJ [8/8]**
>
> **References**
>
> [1] Yuan Zang et al. Word-level Textual Adversarial Attacking as Combinatorial Optimization ACL 2020
>
> [2] Shuhuai Ren et al. Generating Natural Language Adversarial Examples through Probability Weighted Word Saliency, ACL 2019
>
> [3] Miao Xiong et al, Can LLMs Express Their Uncertainty? An Empirical Evaluation of Confidence Elicitation in LLMs, ICLR 2024
>
> [4] Fanghua Ye et al, Benchmarking LLMs via Uncertainty Quantification 2024
>
> [5] Jiuhai Chen et al. Quantifying Uncertainty in Answers from any Language Model and Enhancing their Trustworthiness, ACL 2024
>
> [6] Xuezhi Wang et al. self-consistency improves chain of thought reasoning in language models, ICLR 2023
>
> [7] Saurav Kadavath et al. Language Models (Mostly) Know What They Know, Anthropic technical report 2022
>
> [8] Yusuf Sale et al. Second-Order Uncertainty Quantification: A Distance-Based Approach, ICML 2024
>
> [9] Thomas Savage et al. Large Language Model Uncertainty Measurement and Calibration for Medical Diagnosis and Treatment, 2024
>
> [10] Narmin Ghaffari Laleh et al. Adversarial attacks and adversarial robustness in computational pathology, Nature Communications 2022
>
> [11] Burrell, J. et al. How the machine ‘thinks’: Understanding opacity in machine learning algorithms. Big Data & Society, 2016
>
> [12] Changchang Zeng et al. A Survey on Machine Reading Comprehension: Tasks, Evaluation Metrics and Benchmark Datasets, 2020
>
> [13] Danqi Chen, neural reading comprehension and beyond , 2018
>
> [14] Tomáš Kočiský et al. The NarrativeQA Reading Comprehension Challenge, TACL 2017
>
> [15] Mandar Joshi et al. TriviaQA: A Large Scale Distantly Supervised Challenge Dataset for Reading Comprehension, ACL 2017
>
> [16] Linyu Liu et al. Uncertainty Estimation and Quantification for LLMs: A Simple Supervised Approach ,2024,
>
> [17] Arslan Chaudhry et al. Finetuning Language Models to Emit Linguistic Expressions of Uncertainty , Google Deepmind 2024
>
> [18] It's Morphin' Time! Combating Linguistic Discrimination with Inflectional Perturbations, ACL 2020
>
> [19] Katherine Tian et al. Just Ask for Calibration: Strategies for Eliciting Calibrated Confidence Scores from Language Models Fine-Tuned with Human Feedback EMNLP 2023
>
> [20] Patrick Chao et al. PAIR Jailbreaking Black Box Large Language Models in Twenty Queries, NIPS workshop 2024
>
> [21] Anay Mehrotra et al. Tree of Attacks: Jailbreaking Black-Box LLMs Automatically, NIPS 2024
>
> [22] Haixia Han et al. Enhancing Confidence Expression in Large Language Models Through Learning from Past Experience, 2024
>
> [23] Minhao Cheng et al. Seq2Sick: Evaluating the Robustness of Sequence-to-Sequence Models with Adversarial Examples, AAAI 2020

---

> ### Comment · Reviewer_dcYJ · 2024-11-30
> **Response to authors**
>
> Thank you for your detailed and thoughtful response. I appreciate the additional experiments you have conducted to address the concerns raised, particularly regarding the generalizability of confidence values. The expanded empirical evaluation helps strengthen the argument.
>
> In response to the lack of detail on the ECE calculation, I note that you have provided additional explanations and formulas, which offer a clearer and more comprehensive understanding of the methodology employed to ensure the calibration of the models.
>
> Regarding the complexity and efficiency of the synonym replacement method, I acknowledge the implementation of the new word ranking strategy and the associated experimental validation. The deletion strategy achieves linear complexity by evaluating each word once to assess its impact on the output probability. The resulting improvement in performance, as evidenced by the increase in ASR, represents a meaningful step forward.
>
> The experiments on confidence elicitation with GPT-4 and Claude also show the effectiveness of the attack vector. It is noteworthy that confidence elicitation is a relatively simple process to implement, which may increase its overall impact.
>
> Finally, while the theoretical analysis of extending the attack to text generation tasks is appreciated, more empirical experiments are necessary to further validate this approach. I look forward to the presentation of further empirical results on this aspect in future research.

---

> > ### Author Response · Authors · 2024-12-02
> > **Thank you**
> >
> > We thank the reviewer for taking the time to read our response. We're glad we managed to address your points.

---

### Author Response · Authors · 2024-11-27
**Thank you for the feedback**

We thank the reviewers for the high quality feedback, it has helped us a lot to improve the paper.

We have marked the changes in blue within the document.

Naturaly, If you still have any more questions feel free to discuss them, we are glad to hear your feedback and futher improve the paper.

Best, Authors of ICLR Submission 8520

---

### Meta-Review · Area_Chair_ryBL · 2024-12-21

**Metareview:**

This paper studies an attack on black-box LLMs via leveraging their ability to express confidence. The authors made significant efforts to address several major issues like illustrating adversarial attack on LLMs, real-world attack, more results, etc. After rebuttal, this paper has three positive reviews and one negative review. The only negative review includes three major questions. After reading all comments and rebuttals, I think the authors well addressed two of them and the third one can be addressed by revising the paper. The reviewer did not response or modify the score. In summary, I think this paper can be accepted.

**Additional Comments On Reviewer Discussion:**

Several key issues in the review are summarized and listed as follows.

1. LLM and classification tasks (Reviewer dcYJ, 7pp6)
The authors did give a lot of explanation to address this issue since LLM’s main highlight is generation instead of classification. After rebuttal, no reviewers insist on this point. However, although not a reviewer, I still feel it is inappropriate to continue using adversarial attack to describe attacks on auto-regressive LLMs. After all, no matter what kind of techniques the attackers focus on (e.g. confidence), the goal remains similar to jailbreak or other attacks to manipulate LLMs’ answers. So, insisting on using the term of adversarial attack will cause unnecessary misunderstanding. After all, this is just a suggestion to revise the description of a paper and I am aware of arguing which term to use is a bit boring and out of scope considering an AC’s job. It’s up to you to insist or modify.

2. More results and attacking mainstream black-box LLMs (Reviewer mH4K, dcYJ)
The authors made significant revisions by adding results of more models, more baselines, and SOTA models like GPT-4o. After rebuttal, no reviewers insisted on lacking results.

3. Attack cost (Reviewer dcYJ, Xe6F)
Also, the authors measured the cost in terms of time and give the results during rebuttal. The reviewers acknowledge the results and no further questions are proposed.

---

### Decision · Program_Chairs · 2025-01-22

Accept (Poster)